# Quantifying the relationship between genetic diversity and population size suggests natural selection cannot explain Lewontin's Paradox

**Vince Buffalo***

Institute for Ecology and Evolution, University of Oregon, Eugene, United States

**Abstract** Neutral theory predicts that genetic diversity increases with population size, yet observed levels of diversity across metazoans vary only two orders of magnitude while population sizes vary over several. This unexpectedly narrow range of diversity is known as Lewontin's Paradox of Variation (1974). While some have suggested selection constrains diversity, tests of this hypothesis seem to fall short. Here, I revisit Lewontin's Paradox to assess whether current models of linked selection are capable of reducing diversity to this extent. To quantify the discrepancy between pairwise diversity and census population sizes across species, I combine previously-published estimates of pairwise diversity from 172 metazoan taxa with newly derived estimates of census sizes. Using phylogenetic comparative methods, I show this relationship is significant accounting for phylogeny, but with high phylogenetic signal and evidence that some lineages experience shifts in the evolutionary rate of diversity deep in the past. Additionally, I find a negative relationship between recombination map length and census size, suggesting abundant species have less recombination and experience greater reductions in diversity due to linked selection. However, I show that even assuming strong and abundant selection, models of linked selection are unlikely to explain the observed relationship between diversity and census sizes across species.

***For correspondence:**
vsbuffalo@gmail.com

**Competing interests:** The author declares that no competing interests exist.

## Introduction

A longstanding mystery in evolutionary genetics is that the observed levels of genetic variation across sexual species span an unexpectedly narrow range. Under neutral theory, the average number of nucleotide differences between sequences (pairwise diversity, $\pi$) is determined by the balance of new mutations and their loss by genetic drift (*Kimura and Crow, 1964*; *Malécot, 1948*; *Wright, 1931*). In particular, expected pairwise diversity at neutral sites in a panmictic population of $N_c$ diploids is $\pi \approx 4N_c\mu$, where $\mu$ is the per basepair per generation mutation rate. Given that metazoan germline mutation rates only differ 10-fold ($10^{-8}$–$10^{-9}$, *Kondrashov and Kondrashov, 2010*; *Lynch, 2010*), and census sizes vary over several orders of magnitude, under neutral theory one would expect that pairwise diversity also vary over several orders of magnitude. However, early allozyme surveys revealed that diversity levels across a wide range of species varied just an order of magnitude (*Lewontin, 1974*, p. 208); this is known as Lewontin's ''Paradox of Variation'. With modern sequencing-based estimates of $\pi$ across taxa ranging over only three orders of magnitude (0.01–10%, *Leffler et al., 2012*), Lewontin's paradox remains unresolved through the genomics era.

Early on, explanations for Lewontin's Paradox have been framed in terms of the neutralist–selectionist controversy (*Lewontin, 1974*; *Kimura, 1984*; *Gillespie, 1991*; *Gillespie, 2001*). The neutralist view is that beneficial alleles are sufficiently rare and deleterious alleles are removed sufficiently quickly, that levels of genetic diversity are shaped predominantly by genetic drift and mutation

(*Kimura, 1984*). Specifically, *non-selective* processes decouple the effective population size implied by observed levels of diversity $\widehat{\pi}$, $\widetilde{N}_e = \widehat{\pi}/4\mu$, from the census size, $N_c$. By contrast, the selectionist view is that direct selection and the indirect effects of selection on linked neutral diversity suppress diversity levels across taxa, specifically because the impact of linked selection is greater in large populations. Undoubtedly, these opposing views represent a false dichotomy, as population genomic studies have uncovered evidence for the substantial impact of both demographic history (e.g. *Zhao et al., 2013*; *Palkopoulou et al., 2015*) and linked selection on genome-wide diversity (e.g. *Elyashiv et al., 2016*; *Begun and Aquadro, 1992*; *Aguade et al., 1989*; *McVicker et al., 2009*).

## Possible resolutions of Lewontin's Paradox

A resolution of Lewontin's Paradox would involve a mechanistic description and quantification of the evolutionary processes that prevent diversity from scaling with census sizes across species. This would necessarily connect to the broader literature on the empirical relationship between diversity and population size (*Frankham, 1996*; *Nei and Graur, 1984*; *Soulé, 1976*; *Leroy et al., 2021*), and the ecological and life history correlates of genetic diversity (*Nevo, 1978*; *Powell, 1975*; *Nevo et al., 1984*). Three categories of processes stand out as potentially capable of decoupling census sizes from diversity: non-equilibrium demography, variance and skew in reproductive success, and selective processes.

It has long been appreciated that effective population sizes are typically less than census population sizes, tracing back to early debates between R.A. Fisher and Sewall Wright (*Fisher and Ford, 1947*; *Wright, 1948*). Possible causes of this divergence between effective and census population sizes include demographic history (e.g. population bottlenecks), extinction and recolonization dynamics, or the breeding structure of populations (e.g. the variance in reproductive success and population substructure). Early explanations for Lewontin's Paradox suggested bottlenecks during the last glacial maximum severely reduced population sizes (*Kimura, 1984*; *Ohta and Kimura, 1973*; *Nei and Graur, 1984*), and emphasized that large populations recover to equilibrium diversity levels more slowly (*Nei and Graur, 1984*, *Kimura, 1984* p. 203–204). Another explanation is that cosmopolitan species repeatedly endure extinction and recolonization events, which reduces effective population size (*Maruyama and Kimura, 1980*; *Slatkin, 1977*).

While intermittent demographic events like bottlenecks and recent expansions have long-term impacts on diversity (since mutation-drift equilibrium is reached on the order of size of the population), characteristics of the breeding structure such as high variance ($V_w$) or skew in reproductive success continuously suppress diversity below the levels predicted by the census size (*Wright, 1938*). For example, in many marine animals, females are highly fecund, and dispersing larvae face extremely low survivorship, leading to high variance in reproductive success (*Waples et al., 2018*; *Waples et al., 2013*; *Hedgecock and Pudovkin, 2011*; *Hauser and Carvalho, 2008*). Such ''sweepstakes' reproductive systems can lead to remarkably small ratios of effective to census population size (e.g. $N_e/N_c$ can range from $10^{-6}$–$10^{-2}$), since $N_e/N \approx 1/V_w$ (*Hedgecock, 1994*; *Wright, 1938*; *Nunney, 1993*), and require multiple-merger coalescent processes to describe their genealogies (*Eldon and Wakeley, 2006*). Overall, these reproductive systems diminish the diversity in some species, but seem unlikely to explain Lewontin's Paradox broadly across metazoans.

Alternatively, selective processes, and in particular the indirect effects of selection on linked neutral variation, could potentially explain the observed narrow range of diversity. The earliest mathematical model of hitchhiking was proffered as a explanation of Lewontin's Paradox (*Smith and Haigh, 1974*). Since, linked selection has been shown to impact diversity levels in a variety of species, as evidenced by the correlation between recombination and diversity (*Aguade et al., 1989*; *Begun and Aquadro, 1992*; *Cutter and Payseur, 2003*; *Stephan and Langley, 1998*; *Cai et al., 2009*). Theoretic work to explain this pattern has considered the impact of a steady influx of beneficial mutations (recurrent hitchhiking; *Stephan et al., 1992*; *Stephan, 1995*), and purifying selection against deleterious mutations (background selection, BGS; *Charlesworth et al., 1993*; *Nordborg et al., 1996*; *Hudson and Kaplan, 1994*). Indeed, empirical work indicates background selection diminishes diversity around genic regions in a variety of species (*McVicker et al., 2009*; *Hernandez et al., 2011*; *Charlesworth, 1996*), and now efforts have shifted towards teasing apart the effects of positive and negative selection on genomic diversity (*Elyashiv et al., 2016*).

A class of models that are of particular interest in the context of Lewontin's Paradox are recurrent hitchhiking models that decouple diversity from the census population size. These models predict diversity levels when strongly selected beneficial mutations regularly enter and sweep through the population, trapping lineages and forcing them to coalesce (*Kaplan et al., 1989*; *Gillespie, 2000*). In general, decoupling occurs under these hitchhiking models when the rate of coalescence due to selection is much greater than the rate of neutral coalescence (e.g. *Coop and Ralph, 2012*, Equation 22). In contrast, under other linked selection models, the resulting effective population size is proportional to population size; these models cannot decouple diversity, all else equal. For example, models of background selection and polygenic fitness variation predict diversity is proportional to population size, mediated by the total recombination map length and the deleterious mutation rate or fitness variation (*Charlesworth et al., 1993*; *Nicolaisen and Desai, 2012*; *Nordborg et al., 1996*; *Robertson, 1961*; *Santiago and Caballero, 1995*).

## Recent approaches towards resolving Lewontin's Paradox

Recently, *Corbett-Detig et al., 2015* used population genomic data to estimate the reduction in diversity due to background selection and hitchhiking across 40 species, and showed that the impact of selection increases with two proxies of census population size, species range and the inverse of body size. Based on this evidence, they argued that selection could explain Lewontin's Paradox; however, in a re-analysis, *Coop, 2016* demonstrated that the observed magnitude of these reductions is insufficient to explain the orders-of-magnitude shortfall between observed and expected levels of diversity across species. Other recent work has found that life history characteristics related to parental investment, such as propagule size, are good predictors diversity in animals (*Romiguier et al., 2014*; *Chen et al., 2017*). Nevertheless, while these diversity correlates are important clues, they do not propose a mechanism by which these traits act to constrain diversity within a few orders of magnitude.

Here, I revisit Lewontin's Paradox by integrating several data sets in order to compare the observed relationship between diversity and census size with the predicted relationship under different selection models. Prior surveys of genetic diversity either lacked census population size estimates, used allozyme-based measures of heterozygosity, or included fewer species. To address these shortcomings, I first estimate census sizes by combining predictions of population density based on body size with ranges estimated from geographic occurrence data. Using these estimates, I quantify the relationship between census size and previously-published genomic diversity estimates across 172 metazoan taxa within nine phyla, thus characterizing the relationship between $\pi$ and $N_c$ that underlies Lewontin's Paradox.

Past work looking at the relationship between $\pi$ and $N_c$ has been unable to fully account for phylogenetic non-independence across taxa (*Felsenstein, 1985*). To address this, I use phylogenetic comparative methods (PCMs) with a synthetic time-calibrated phylogeny to account for shared phylogenetic history. Moreover, it is disputed whether considering phylogenetic non-independence is necessary in population genetics, given that coalescent times within species are much less than divergence times (*Whitney and Garland, 2010*; *Lynch, 2011*). Using PCMs, I address this by estimating the degree of phylogenetic signal in the diversity census size relationship, and investigating how these traits evolve along the phylogeny.

Finally, I explore whether the predicted reductions of diversity under background selection and recurrent hitchhiking are sufficiently strong to resolve Lewontin's Paradox. I do so using selection parameters from *Drosophila melanogaster*, a species known to be strongly affected by linked selection. Given the effects of linked selection are mediated by recombination map length, I also investigate how map lengths vary with census population size using data from a previously-published survey (*Stapley et al., 2017*). I find map lengths are typically shorter in large–census-size species, increasing the effects of linked selection in these species, which could further decouple diversity from census size. Still, I find the combined impact of these modes of linked selection fall short in explaining Lewontin's Paradox, and discuss future avenues through which the Paradox of Variation could be fully resolved.

## Results

### Estimates of census population size

An impediment in resolving Lewontin's Paradox is characterizing the relationship between diversity and census population sizes. This is difficult because census population sizes are unavailable for many taxa, especially for extremely abundant, cosmopolitan species that define the upper limit of ranges. Previous work has surveyed the literature for census size estimates (*Nei and Graur, 1984*; *Soulé, 1976*; *Frankham, 1996*), or used range, body size, or qualitative categories as proxies for census size (*Corbett-Detig et al., 2015*; *Leffler et al., 2012*). To quantify the relationship between genomic estimates of diversity and census population sizes, I first approximate census population sizes for 172 metazoan taxa (*Figure 1*). I estimate population densities based on an empirical linear relationship between body sizes and density that holds across metazoans (see *Figure 1—figure supplement 1*; *Damuth, 1981*; *Damuth, 1987*). Then, from geographic occurrence data, I estimate range sizes. Finally, I estimate population size as the product of these predicted densities and range estimates (see Materials and methods: Macroecological Estimates of Population Size). Note that the relationship between population density and body size is driven by energy budgets, and thus reflects macroecological equilibria (*Damuth, 1987*). Consequently, population sizes are underestimated for taxa like humans and their domesticated species, and overestimated for species with anthropogenically-reduced densities or fragmented ranges. For example, the population size of *Lynx lynx* is likely around 50,000 (*IUCN, 2020*) which is around two orders of magnitude smaller than my estimate. Additionally, the range size estimates do not consider whether an area has unsuitable habitat, and thus may be overestimated for species with particular niches or patchy habitats. While my approach produces approximate and sometimes crude estimates, it has the advantage that it can be efficiently calculated for numerous taxa, which is sufficient to estimate the magnitude of Lewontin's Paradox (see Population Size Validation for more on validation based on biomass and other approaches).

### Characterizing the Diversity–Census-size Relationship

To determine which ecological or evolutionary processes could decouple diversity from census population size, we first need to quantify this relationship across a wide variety of taxa. Previous work has found there is a significant relationship between heterozygosity and the logarithm of population size or range size, but these studies relied on heterozygosity measured from allozyme data (*Soulé, 1976*; *Frankham, 1996*; *Nei and Graur, 1984*). I confirm these findings using pairwise diversity estimates from genomic sequence data and the estimated census sizes (*Figure 2*). The pairwise diversity estimates are from three sources: *Leffler et al., 2012*, *Corbett-Detig et al., 2015*, and *Romiguier et al., 2014*, and are predominantly from either synonymous or non-coding DNA (see Methods and Materials: 4.1 Diversity and Map Length Data). Overall, an ordinary least squares (OLS) relationship on a log-log scale fits the data well (*Figure 2*, gray dashed line). The OLS slope estimate is significant and implies a 13% percent increase in differences per basepair for every order of magnitude census size grows (95% confidence interval [12%, 14%], adjusted $R^2 = 0.26$; see also the OLS fit per-phyla, *Figure 2—figure supplement 2*).

Notably, this relationship has few outliers and is relatively homoscedastic. This is in part because of the log-log scale, in contrast to previous work (*Nei and Graur, 1984*; *Soulé, 1976*); see *Figure 2—figure supplement 1* for a version on a log-linear scale. However, it is noteworthy that few taxa have diversity estimates below $10^{-3.5}$ differences per basepair. Those that do, lynx (*Lynx Lynx*), wolverine (*Gulo gulo*), and Massasauga rattlesnake (*Sistrurus catenatus*) face habitat loss and declining population sizes. These three species are all in the IUCN Red List, but are listed as least concern (though their presence in the Red List indicates they are of conservation interest). In Appendix D, Appendix D Diversity and IUCN Red List Status, I explore the relationships between IUCN Red List status, diversity, and population size.

### Phylogenetic non-independence and the population size diversity relationship

One limitation of using ordinary least squares is that shared phylogenetic history can create correlation structure in the residuals, which violates an assumption of the regression model and can lead to bias (*Felsenstein, 1985*; *Revell, 2010*). To address this shortcoming, I fit the diversity–census-size

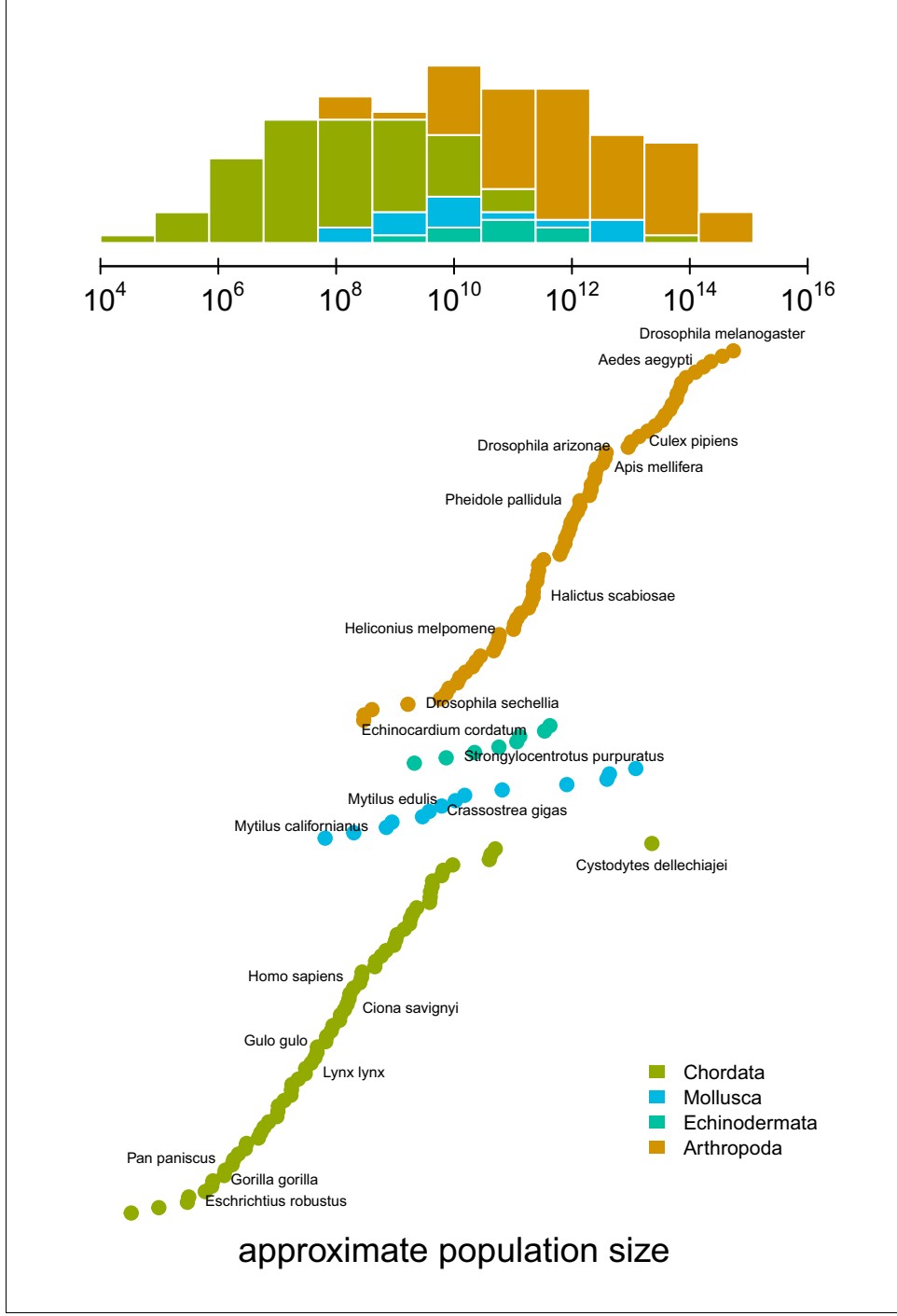

**Figure 1.** The distribution of approximate census population sizes estimated by this study. Some phyla containing few species were excluded for clarity.

The online version of this article includes the following source data and figure supplement(s) for figure 1:

**Source data 1.** The population size estimates for 172 metazoan taxa.

**Figure supplement 1.** The relationship between body mass and population density found by *Damuth, 1987*, which is used to predict population densities.

**Figure supplement 2.** The fraction of total species per class on earth included in this study's sample, per class.

**Figure supplement 3.** Comparison of this paper's range estimates procedure against the IUCN Red List's range estimates.

**Figure supplement 4.** Validation of this paper's range estimates against the categorical labels of *Leffler et al., 2012*.

*Figure 1 continued on next page*

*Figure 1 continued*

**Figure supplement 5.** The relationship between body length (meters) and body mass (grams) in the *Romiguier et al., 2014* data set.

relationship using a phylogenetic mixed-effects model, investigated whether there is a signal of phylogenetic non-independence, estimated the continuous trait values on the phylogeny, and explored how diversity and population size evolve. Prior population genetic comparative studies have lacked time-calibrated phylogenies and assumed unit branch lengths (*Whitney and Garland, 2010*), a

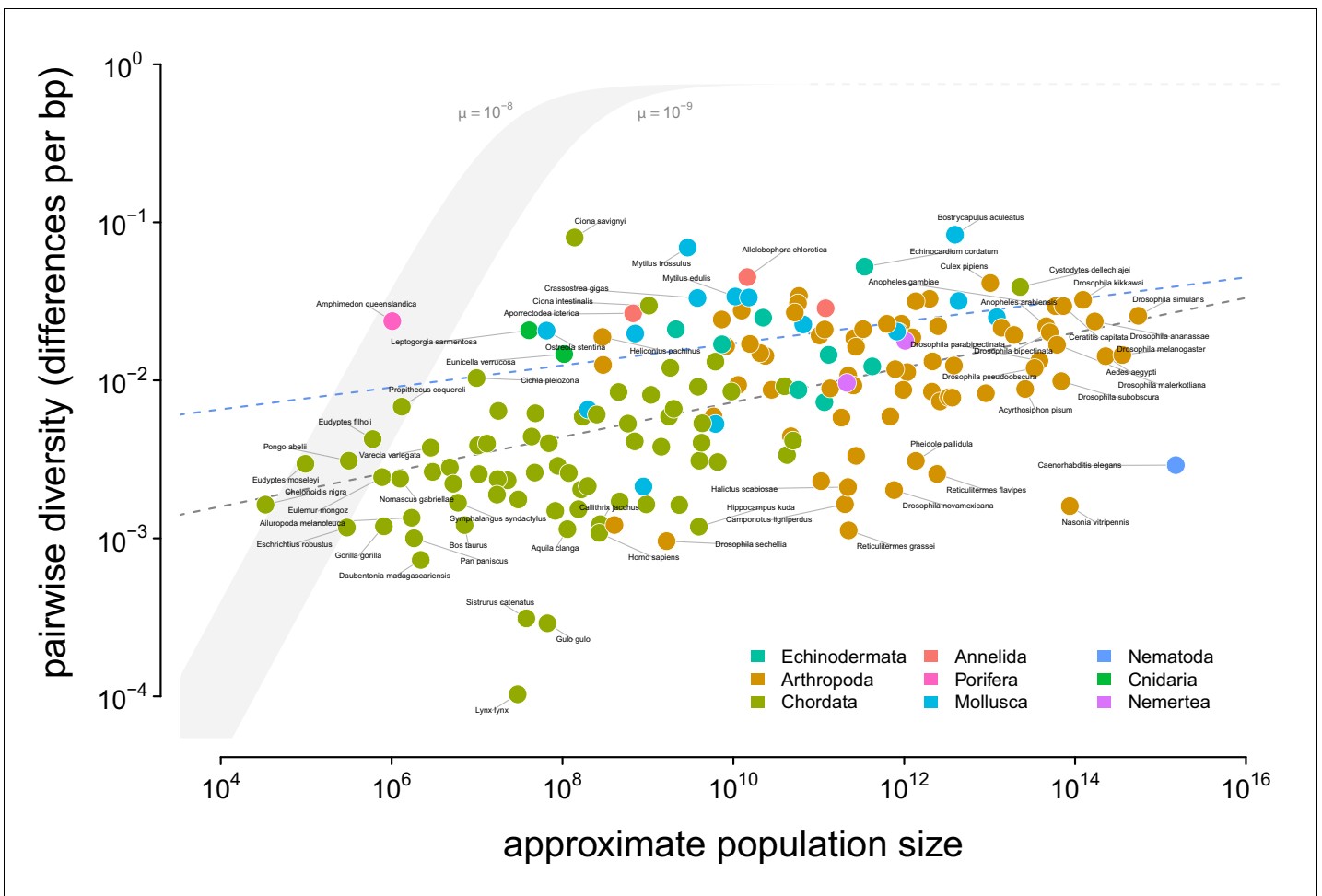

**Figure 2.** A visualization of Lewontin's Paradox of Variation. Pairwise diversity (data from *Leffler et al., 2012*, *Corbett-Detig et al., 2015*, and *Romiguier et al., 2014*), which varies over three orders of magnitude, shows a weak relationship with approximate population size, which varies over 12 orders of magnitude. The shaded curve shows the range of expected neutral diversity if $N_e$ were to equal $N_c$ under the four-alleles model, $\log_{10}(\pi) = \log_{10}(\theta) - \log_{10}(1 + 4\theta/3)$ where $\theta = 4N_c\mu$, for two mutation rates, $\mu = 10^{-8}$ and $\mu = 10^{-9}$, and the light gray dashed line represents the maximum pairwise diversity under the four alleles model. The dark gray dashed line is the OLS regression fit, and the blue dashed line is the regression fit using a phylogenetic mixed-effects model. Points are colored by phylum. The species *Equus ferus przewalskii* ($N_c \approx 10^3$ and $\pi = 3.6 \times 10^{-3}$) was an outlier and excluded from this figure for visual clarity.

The online version of this article includes the following source data and figure supplement(s) for figure 2:

**Source data 1.** The diversity and population size dataset for 172 metazoan taxa.
**Figure supplement 1.** A linear-log version of *Figure 2*.
**Figure supplement 2.** A version of *Figure 2* with OLS estimates per phylum.
**Figure supplement 3.** The posterior distributions and fitted relationship between diversity and both body mass and range size.
**Figure supplement 4.** Pairwise diversity grouped by the range categories from *Leffler et al., 2012*, with point size indicating the predicted population density.

shortcoming that has drawn criticism (*Lynch, 2011*). I use a synthetic time-calibrated phylogeny created from the DateLife project (*O'Meara et al., 2020*) to account for shared phylogenetic history (see Materials and methods: Phylogenetic Comparative Methods).

Using a phylogenetic mixed-effects model (*Lynch, 1991*; *Hadfield and Nakagawa, 2010*; *de Villemereuil and Nakagawa, 2014*) implemented in Stan (*Carpenter et al., 2017*; *Stan Development Team, 2020*), I estimated the linear relationship between diversity and population size (on a log-log scale) accounting for phylogeny, for the 166 taxa without missing data and present in the synthetic chronogram. This type of model is needed because closely-related species may differ from the average trend between $N_c$ and $\pi$ in similar ways due to shared phylogenetic history, similar life history traits, etc., and thus do not represent independent observations as is assumed by the standard regression model. This is a form of phylogenetic pseudoreplication, and can be accounted for with a phylogenetic mixed-effects model. The phylogenetic mixed-effects model does not assume that there is phylogenetic structure in either $N_c$ or $\pi$ (which itself is not a violation of the standard regression model, *Revell, 2010* and *Uyeda et al., 2018*), but rather accounts

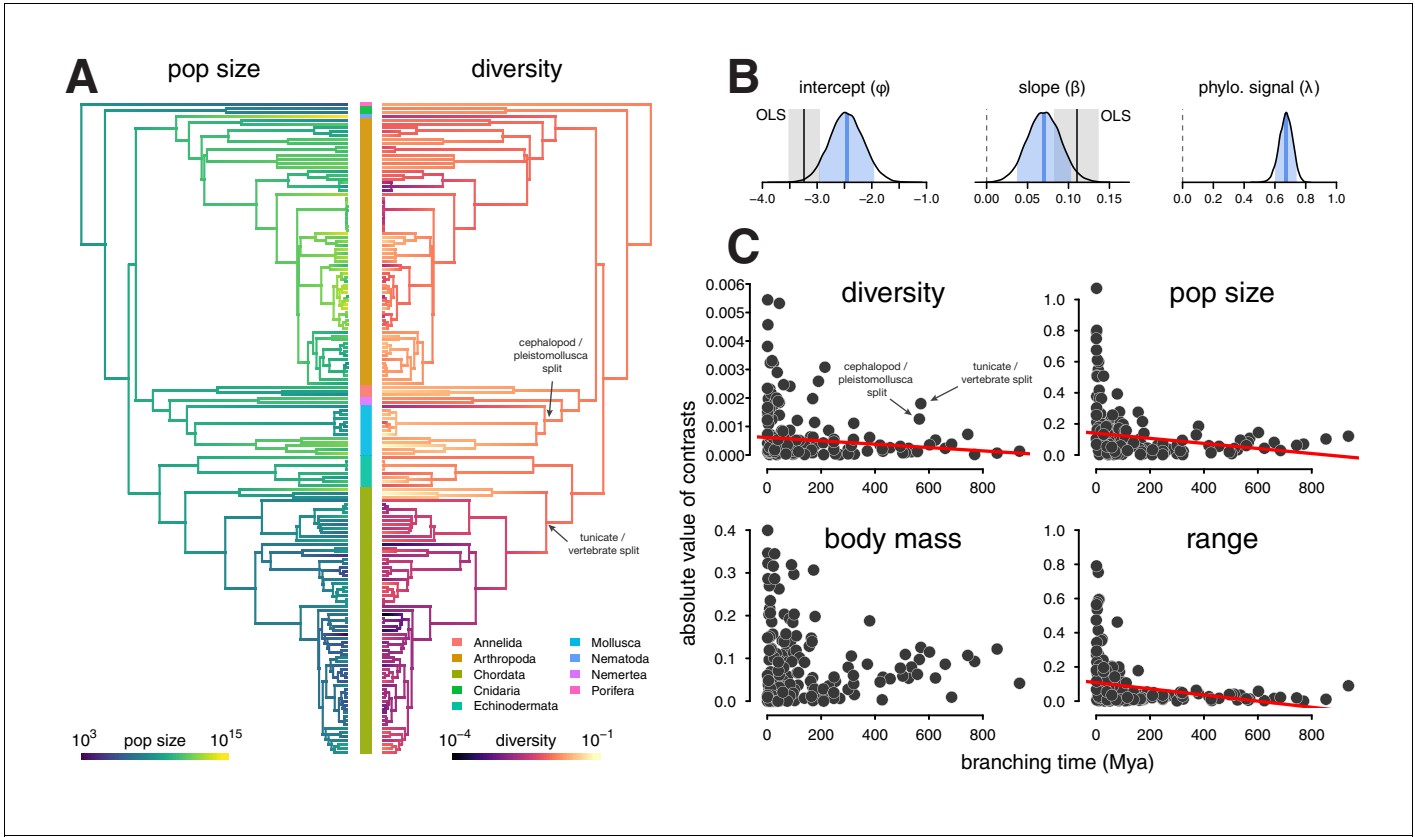

**Figure 3.** Phylogenetic comparative models of diversity and population size. (**A**) The ancestral continuous trait estimates for the population size and diversity (differences per bp, log scaled) across the phylogeny of 166 taxa. The phyla of the tips are indicated by the color bar in the center. (**B**) The posterior distributions of the intercept, slope, and phylogenetic signal ($\lambda$, *de Villemereuil and Nakagawa, 2014*) of the phylogenetic mixed-effects model of diversity and population size (log scaled). Also shown are the 90% credible interval (light blue shading), posterior mean (blue line), OLS estimate (gray solid line), and bootstrap OLS confidence intervals (light gray shading). (**C**) The node-height tests of diversity, population size, and the two components of the population size estimates, body mass, and range (all traits on log scale before contrast was calculated). Each point shows the standardized phylogenetic independent contrast and branching time for a pair of lineages. Red lines are robust regression estimates (and are only shown for statistically significant relationships at the $\alpha = 0.05$ level). Note that some outlier pairs with very high phylogenetic independent contrasts were excluded (in all cases, these outliers were in the genus *Drosophila*).

The online version of this article includes the following figure supplement(s) for figure 3:

**Figure supplement 1.** The posterior distributions for the parameters of the phylogenetic mixed-effects model of diversity and population size (this is analogous to *Figure 3B*) fit separately on chordates ($n = 68$), molluscs ($n = 13$), and arthropods ($n = 68$).

**Figure supplement 2.** The ancestral continuous trait estimates for diversity and population size with species labels.

**Figure supplement 3.** The ancestral continuous trait estimates for recombination map length and diversity and population size with species labels.

for phylogenetic correlation structure in the residuals if any is present. Importantly, phylogenetic mixed-effects models simultaneously estimate the degree of phylogenetic structure in the residuals while fitting the relationship between $N_c$ and $\pi$. If the residuals are distributed independently, the estimated relationship would be similar to that found by ordinary least squares, and the estimated phylogenetic signal would be zero. Overall, this approach is conservative, making no assumptions about the source of the phylogenetic signal while accounting for violations of the regression model due to dependence among the residuals if present (see *Revell, 2010* for a discussion of this).

As with the linear regression, I find this relationship is positive and significant (95% credible interval 0.03, 0.11), though somewhat attenuated compared to the OLS estimates (*Figure 3B*). Since the population size estimates are based on range and body mass, they are essentially a composite trait; fitting phylogenetic mixed-effects models separately on body mass and range indicates these have significant positive and negative effects, respectively (*Figure 2—figure supplement 3*; see also *Figure 2—figure supplement 4* for the relationship between diversity and the range categories of *Leffler et al., 2012*).

Since the phylogenetic mixed-effects model simultaneously estimates the variance of the phylogenetic effect ($\sigma_p^2$) and the residual variance ($\sigma_r^2$), these can be used to estimate the phylogenetic signal, $\lambda = \sigma_p^2/(\sigma_p^2 + \sigma_r^2)$ (*Lynch, 1991*; *de Villemereuil and Nakagawa, 2014*; see *Freckleton et al., 2002* for a comparison to Pagel's $\lambda$). When residuals are free of correlations due to shared phylogenetic history, then $\lambda = 0$ and all the variance could be explained by evolution or noise on the tips. In the relationship between population size and diversity, the posterior mean of $\lambda = 0.67$ (90% credible interval $[0.58, 0.75]$) indicates a majority of the variance perhaps might be due to shared phylogenetic history (*Figure 3B*).

This high degree of phylogenetic signal substantiates Gillespie's concern (*Gillespie, 1991*) that the $\pi$–$N_c$ relationship may be driven by chordate-arthropod differences. A visual inspection of the estimated ancestral continuous values for diversity and population size on the phylogeny indicates the high phylogenetic signal seems to be driven in part by chordates having low diversity and small population sizes compared to non-chordates (*Figure 3A*). This problem resembles Felsenstein's worst-case scenario (*Felsenstein, 1985*; *Uyeda et al., 2018*), where a singular event on a lineage separating two clades generates a spurious association between two traits.

To investigate whether clade-level differences dominated the relationship between diversity and population size, I fit phylogenetic mixed-effects models to phyla-level subsets of the data for clades with sufficient sample sizes (see Methods: 4.4 Phylogenetic Comparative Methods). This analysis shows a significant positive relationship between diversity and population size in arthropods, and positive weak relationships in molluscs and chordates (*Figure 3—figure supplement 1*). Each of the 90% credible intervals for slope overlap, suggesting the relationship between $\pi$ and $N_c$ is similar across these clades.

Additionally, I have explored the rate of trait change through time using node-height tests (*Freckleton and Harvey, 2006*). Node-height tests regress the absolute values of the standardized contrasts between lineages against the branching time (since present) of these lineages. Under Brownian Motion (BM), standardized contrasts are estimates of the rate of character evolution (*Felsenstein, 1985*); if a trait evolves under constant rate BM, this relationship should be flat. For both diversity and population size, node-height tests indicate a significant increase in the rate of evolution towards the present (robust regression p-values 0.023 and 0.00018 respectively; *Figure 3C*). Considering the constituents of the population size estimate, range and body mass, separately, the rate of evolution of range but not body mass shows a significant increase (p-value $1.03 \times 10^{-7}$) towards the present.

Interestingly, the diversity node-height test reveals two rate shifts at deeper splits (*Figure 3C*, top left) around 570 Mya. These nodes represent the branches between tunicates and vertebrates in chordates, and cephalopods and pleistomollusca (bivalves and gastropods) in molluscs. While the cephalopod-pleistomollusca split outlier may be an artifact of having a single cephalopod (*Sepia officinalis*) in the phylogeny, the tunicate-vertebrate split outlier is driven by the low diversity of vertebrates and the previously-documented exceptionally high diversity of tunicates (sea squirts; *Nydam and Harrison, 2010*; *Small et al., 2007*). This deep node representing a rate shift in diversity could reflect a change in either effective population size or mutation rate, and there is some evidence of both in this genus *Ciona* (*Small et al., 2007*; *Tsagkogeorga et al., 2012*). Neither of these

deep rate shifts in diversity is mirrored in the population size node-height test (*Figure 3C*, top right). Rather, it appears a trait impacting diversity but not census size (e.g. mutation rate or offspring distributions) has experienced a shift on the lineage separating tunicates and vertebrates. At nearly 600 Mya, these deep nodes illustrate that expected effective population sizes (and thus coalescence times) can share phylogenetic history, due to phylogenetic inertia in some combination of population size, reproductive system, and mutation rates.

Finally, an important caveat is the increase in rate towards the tips could be caused by measurement noise, or possibly uncertainty or bias in the divergence time estimates deep in the tree. Inspecting the lineage pairs that lead to this increase in rate towards the tips indicates these represent plausible rate shifts, e.g. between cosmopolitan and endemic sister species like *Drosophila simulans* and *Drosophila sechellia*; however, ruling out measurement noise entirely as an explanation would involve modeling the uncertainty of diversity and population size estimates.

## Assessing the impact of linked selection on diversity across taxa

The above analyses reemphasize the drastic shortfall of diversity levels as compared to census sizes. Linked selection has been proposed as the mechanism that acts to reduce diversity levels from what we would expect given census sizes (*Smith and Haigh, 1974*; *Gillespie, 2000*; *Corbett-Detig et al., 2015*). Here, I test this hypothesis by estimating the scale of diversity reductions expected under background selection and recurrent hitchhiking, and comparing these to the observed relationship between $\pi$ and $N_c$.

I quantify the effect of linked selection on diversity as the ratio of observed diversity ($\pi$) to the estimated diversity in the absence of linked selection ($\pi_0$), $R = \pi/\pi_0$. Here, $\pi_0$ would reflect only demographic history and non-heritable variation in reproductive success. There are two difficulties in evaluating whether linked selection could resolve Lewontin's Paradox. The first difficulty is that $\pi_0$ is unobserved. Previous work has estimated $\pi_0$ using methods that exploit the spatial heterogeneity in recombination and functional density across the genome to fit linked selection models that incorporate both hitchhiking and background selection (*Elyashiv et al., 2016*; *Corbett-Detig et al., 2015*). The second difficulty is understanding how $R$ varies across taxa, since we lack estimates of critical model parameters for most species. Still, I can address a key question: if diversity levels were determined by census sizes ($\pi_0 = 4N_c\mu$), would the combined effects of background selection and recurrent hitchhiking be sufficient to reduce diversity to observed levels? Furthermore, does the relationship between census size and predicted diversity under linked selection across species, $\pi_{BGS+HH} = R\pi_0$, match the observed relationship in *Figure 2*?

Since we lack estimates of selection parameters across species, I parameterize the hitchhiking and BGS models using estimates from *Drosophila melanogaster*, a species known to be strongly affected by linked selection (*Sella et al., 2009*). Under a generalized model of hitchhiking and background selection (*Elyashiv et al., 2016*; *Coop and Ralph, 2012*) and assuming $N_e = N_c$, the expected diversity is

$$\pi_{\mathrm{BGS+HH}} \approx \frac{\theta}{1/B(U,L) + 2N_c S(\gamma,J,L)} \tag{1}$$

where $\theta = 4N_c\mu$, $B(U,L)$ is the effect of background selection, and $S(\gamma,J,L)$ is the rate of coalescence caused by sweeps (*Elyashiv et al., 2016*, Equation 1, *Coop and Ralph, 2012*, Equation 20). Under background selection models with recombination, the reduction is $B(U,L) = \exp(-U/L)$ where $U$ is the per diploid genome per generation deleterious mutation rate, and $L$ is the recombination map length in Morgans (*Hudson and Kaplan, 1994*; *Nordborg et al., 1996*). This BGS model is similar to models of effective population size under polygenic fitness variation, and can account for other modes of linked selection (*Robertson, 1961*; *Santiago and Caballero, 1995*; *Santiago and Caballero, 1998*, see Appendix 2, Background Selection and Polygenic Fitness Models). The coalescence rate due to sweeps is $S(\gamma,J,L) = \frac{\gamma}{L}J$, where $\gamma$ is the number of adaptive substitutions per generation, and $J$ is the probability a lineage is trapped by sweeps as they occur across the genome ($J_{2,2}$ in Equation 15 of *Coop and Ralph, 2012*).

Parameterizing the model this way, I then set the key parameters that determine the impact of recurrent hitchhiking and background selection ($\gamma$, $J$, and $U$) to strong selection values estimated for *Drosophila melanogaster* by *Elyashiv et al., 2016*. My estimate of the adaptive substitutions per

generation ($\gamma_{\text{Dmel}} \approx 2.3 \times 10^{-3}$) based Elyashiv et al. implies a rate of sweeps per basepair of $\nu_{\text{BP,Dmel}} \approx 2.34 \times 10^{-11}$, which is close to other estimates from *D. melanogaster* (see *Figure 4—figure supplement 5A*). The rate of deleterious mutations per diploid genome, per generation is parameterized using the estimate from Elyashiv et al., $U_{\text{Dmel}} = 1.6$, which is slightly greater than previous estimates based on Bateman-Mukai approaches (*Mukai, 1985*; *Mukai, 1988*; *Charlesworth, 1987*). Finally, the probability that a lineage is trapped in a sweep, $J_{\text{Dmel}} \approx 4.5 \times 10^{-4}$, is calculated from the estimated genome-wide average coalescence rate due to sweeps from Elyashiv et al. (see *Figure 4—figure supplement 5B* and Materials and methods: Predicted Reductions in Diversity for more details on parameter estimates). Using these parameters, I then explore how the predicted range of diversity levels varies across species with recombination map length ($L$) and census population size ($N_c$).

Previous work has found that the impact of linked selection increases with $N_c$ (*Corbett-Detig et al., 2015*; see *Figure 4—figure supplement 4A*), and it is often thought that this is driven by higher rates of adaptive substitutions in larger populations (*Ohta, 1992*), despite equivocal evidence (*Galtier, 2016*). However, there is another mechanism by which species with larger population sizes might experience a greater impact of linked selection: recombination map length, $L$, is known to correlate with body mass (*Burt and Bell, 1987*) and thus varies inversely with population size. As this is a critical parameter that determines the genome-wide impact of both hitchhiking and background selection, I examine the relationship between recombination map length ($L$) and census population size ($N_c$) across taxa, using available estimates of map lengths across species (*Stapley et al., 2017*; *Corbett-Detig et al., 2015*). I find a significant non-linear relationship using phylogenetic mixed-effects models (*Figure 4A*; see Methods and materials: 4.4 Phylogenetic Comparative Methods). There is also a correlation between map length and genome size (*Figure 4—figure supplement 2*) and genome size and population size (*Figure 4—figure supplement 1*). These findings are consistent with the hypothesis that non-adaptive processes increase genome size in small-$N_e$ species (*Lynch and Conery, 2003*) which in turn could increase map lengths, as well as the hypothesis that map lengths are adaptively longer to more efficiently select against deleterious alleles in smaller populations (*Roze, 2021*). Overall, the negative relationship between map length and census size indicates linked selection is expected to be stronger in species with short map lengths, which are high-$N_c$ species.

Then, I predict the expected diversity ($\pi_{BGS+HH}$) under background selection and hitchhiking, assuming $N_e = N_c$ and that all species had the rate of sweeps and strength of BGS as *D. melanogaster*. Since neutral mutation rates μ are unknown and vary across species, I calculate the range of predicted $\pi_{BGS+HH}$ estimates for μ = $10^{-9}$–$10^{-8}$ (using the four-alleles model, *Tajima, 1996*), and compare this to the observed relationship between π and $N_c$ in *Figure 4B*. Under these parameters and assumptions, linked selection begins to appreciably constrain diversity for $N_c \gtrsim 10^7$, since $S(\gamma_{\text{Dmel}}, J_{\text{Dmel}}, L) \approx 10^{-8}$–$10^{-7}$ and linked selection dominates drift when $S(\gamma, J, L) > 1/2N$. Overall, this reveals two problems for the hypothesis that linked selection could solve Lewontin's Paradox. First, low to mid-$N_c$ species (census sizes between $10^4$–$10^7$) have sufficiently long map lengths that their diversity levels are only moderately reduced by linked selection, leading to a wide gap between predicted and observed diversity levels. For this not to be the case, the rate of adaptive mutations or the deleterious mutation rate would need to be orders of magnitude higher for species within this range than in *Drosophila melanogaster*, which is incompatible with the rate of adaptive protein substitutions across species (*Galtier, 2016*) and overall mutation rates (*Lynch, 2010*). Furthermore, linked selection has been quantified in humans, which fall in this census size range, and has been found to be relatively weak (*McVicker et al., 2009*; *Hernandez et al., 2011*; *Hellmann et al., 2008*; *Cai et al., 2009*; *Boyko et al., 2008*). Second, while hitchhiking and BGS can reduce predicted diversity levels for high-$N_c$ species ($N_c > 10^{12}$) to observed levels, this would imply available estimates of $\pi_0$ are underestimated by several orders of magnitude in *Drosophila* (*Figure 4—figure supplement 4B*). The high reductions in π predicted here (compared to those of *Elyashiv et al., 2016*) are a result of using $N_c$, rather than $N_e = \pi_0/4\mu$ in the denominator of *Equation (1)*, which leads to a very high rate of sweeps in the population. I do not consider selective interference, though the saturation of adaptive substitutions per Morgan would only act to limit the reduction in diversity (*Weissman and Barton, 2012*), and thus these results are conservative.

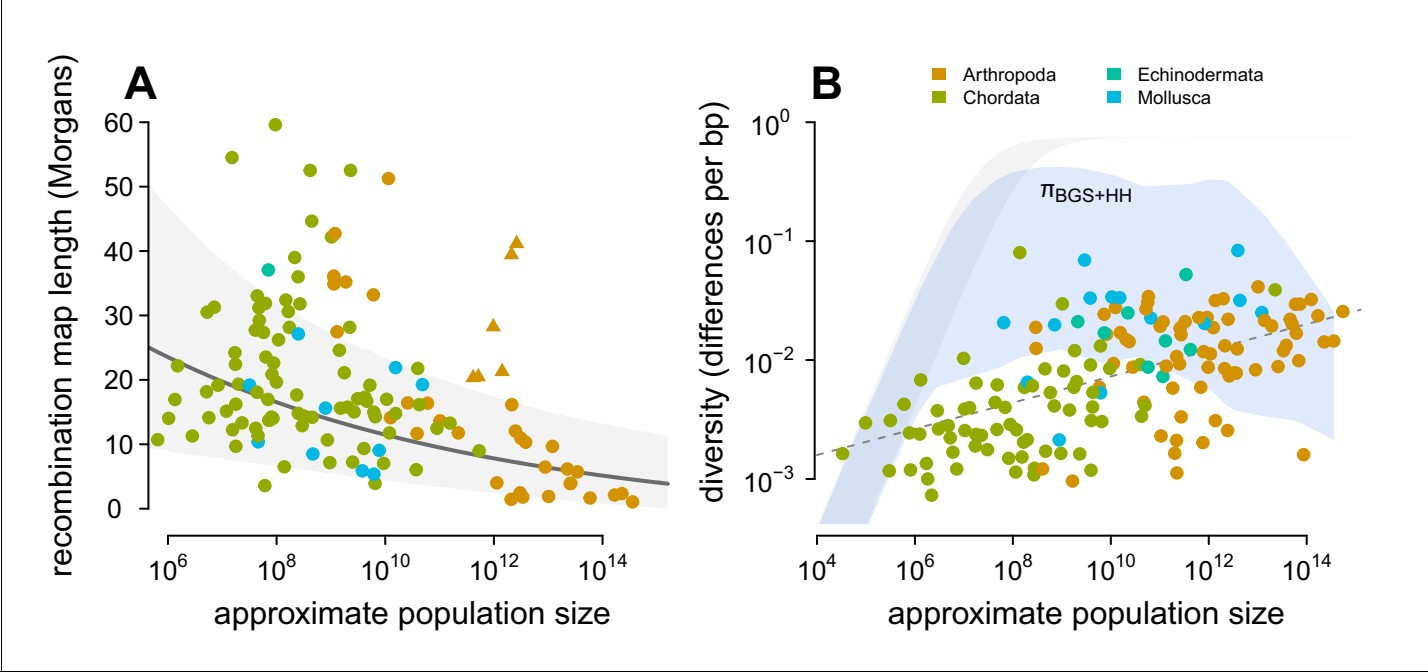

**Figure 4.** Predicting the impact of linked selection on diversity. (A) The observed relationship between recombination map length ($L$) and census size ($N_c$) across 136 species with complete data and known phylogeny. Triangle points indicate six social taxa excluded from the model fitting since these have adaptively higher recombination map lengths (*Wilfert et al., 2007*). The dark gray line is the estimated relationship under a phylogenetic mixed-effects model, and the gray interval is the 95% posterior average. (B) Points indicate the observed $\pi$–$N_c$ relationship across taxa shown in *Figure 2*, and the blue ribbon is the range of predicted diversity were $N_e = N_c$ for $\mu = 10^{-8}$–$10^{-9}$, and after accounting for the expected reduction in diversity due to background selection and recurrent hitchhiking under *Drosophila melanogaster* parameters. In both plots, point color indicates phylum.

The online version of this article includes the following source data and figure supplement(s) for figure 4:

**Source data 1.** The map length, population size, and linked selection estimates for 136 metazoan taxa.

**Figure supplement 1.** The relationship between genome size and approximate census population size.

**Figure supplement 2.** The relationship between genome size and recombination map length.

**Figure supplement 3.** The observed $\pi$–$N_c$ relationship (points) across species compared to the predicted diversity (ribbons) under different modes of linked selection and parameters, for a range of mutation rates, $10^{-9}$–$10^{-8}$.

**Figure supplement 4.** The relationship between $N_c$ and diversity in the *Corbett-Detig et al., 2015* data, and the relationship between estimated reduction in diversity and census size, for three different approaches.

**Figure supplement 5.** Comparison of the *Drosophila* sweep parameters used in this study with parameters from other studies.

Finally, the poor fit between observed and predicted levels of diversity across species is not remedied by stronger selection parameters. In *Figure 4—figure supplement 3B*, I increase both selection parameters $U$ and $\gamma$ ten-fold each, and find the same qualitative pattern: on a log-log scale the relationship between $N_c$ and $\pi$ is linear, while the predicted diversity under linked selection is non-linear with $N_c$. Under this ten-fold higher selection regime, there is more overlap between observed and predicted levels of diversity, but diversity is severely *under-predicted* for high-$N_c$ species. Additionally, this would imply that selection in low-to-mid-$N_c$ species is ten-folder higher than estimated in *Drosophila melanogaster*, which is implausible. Overall, this suggests that present models of linked selection, even with very strong selection across species, are qualitatively incapable of matching the observed relationship between $N_c$ and $\pi$ and thus cannot explain Lewontin's Paradox.

## Discussion

Nearly fifty years after Lewontin's description of the Paradox of Variation, how evolutionary, life history, and ecological processes interact to constrain diversity across taxa to a narrow range remains a mystery. I revisit Lewontin's Paradox by first characterizing the relationship between genomic estimates of pairwise diversity and approximate census population size across 172 metazoan species. Previous surveys have used allozyme-based estimates, fewer taxa, or proxies of population size. My

estimates of census population sizes are rough approximates, since they use body size to predict density. An improved estimate might account for vagility (as *Soulé, 1976* did), though this is harder to do systematically across many taxa. Future work might also use other ecological information, such as total biomass, or species distribution modeling to improve census size estimates (*Bar-On et al., 2018*; *Mora et al., 2011*). Still, it seems more accurate estimates would be unlikely to change the qualitative findings here, which resemble those of early surveys (*Nei and Graur, 1984*; *Soulé, 1976*).

One limitation of this study is that diversity estimates are collated from a variety of sources rather than estimated with a single bioinformatic pipeline. This leads to technical noise across diversity estimates; perhaps the relationship between $\pi$ and $N_c$ found here could be tighter with a standardized bioinformatic pipeline. In addition, there might be systematic bioinformatic sources of bias: for example high-diversity sequences may fail to align to the reference genome and end up unaccounted for, leading to a downward bias. Alternatively, a high-diversity sequences might map to the reference genome, but adjacent mis-matching SNPs might be mistaken for a short insertion or deletion. While these issues might affect estimates in high-diversity species, it is unlikely to change the qualitative relationship between $\pi$–$N_c$.

## Macroevolution and Across-Taxa population genomics

Lewontin's Paradox arises from a comparison of diversity across species, yet it has been disputed whether such comparisons require phylogenetic comparative methods. Extending previous work that has accounted for phylogeny in particular clades (*Leffler et al., 2012*), or using taxonomical-level averages (*Romiguier et al., 2014*), I show that the positive relationship between diversity and census size is significant using a mixed-effects model with a time-calibrated phylogeny. Additionally, I find a high degree of phylogenetic signal, evidence of deep shifts in the rate of evolution of genetic diversity, and that arthropods and chordates form clusters. Overall, this suggests that previous concerns about phylogenetic non-independence in comparative population genetic studies were warranted (*Gillespie, 1991*; *Whitney and Garland, 2010*). Notably, *Lynch, 2011* has argued that PCMs for pairwise diversity are unnecessary, since mutation rate evolution is fast and thus free of phylogenetic inertia, sampling variance should exceed the variance due to phylogenetic shared history, and coalescence times are much shorter than divergence times. Since my findings suggest PCMs are necessary in some cases, it is worthwhile to address these points.

First, Lynch has correctly pointed out that while coalescence times are much less than divergence times and should be free of phylogenetic shared history, the factors that determine coalescence times (e.g. mutation rates and effective population size) may not be (*Lynch, 2011*). In other words, coalescence times are free from phylogenetic shared history *were we to condition* on these causal factors that could be affected by shared phylogenetic history. My estimates of phylogenetic signal in the residuals, by contrast, are not conditioned on these factors. Importantly, even "correcting for" phylogeny implicitly favors certain causal interpretations over others (*Westoby et al., 1995*; *Uyeda et al., 2018*). Future work could try to untangle what causal factors determine coalescence times across species, as well as how these factors evolve across macroevolutionary timescales. Second, it is a misconception that a fast rate of trait evolution necessarily reduces phylogenetic signal (*Revell et al., 2008*), and that if either or both variables in a regression are free of phylogenetic signal, PCMs are unnecessary (*Revell, 2010*; *Uyeda et al., 2018*). The evidence of high phylogenetic signal found in this study suggests PCMs are necessary when fitting the relationship between $N_c$ and $\pi$ in order to account for correlated residuals among closely-related species, and to avoid spurious results from phylogenetic pseudoreplication.

Finally, beyond just accounting for phylogenetic non-independence, macroevolution and phylogenetic comparative methods are a promising way to approach across-species population genomic questions. For example, one could imagine that diversification processes could contribute to Lewontin's Paradox. If large-$N_c$ species were to have a rate of speciation that is greater than the rate at which mutation and drift reach equilibrium (which is indeed slower for large $N_c$ species), this could act to decouple diversity from census population size. That is to say, even if the rate of random demographic bottlenecks were constant across taxa, lineage-specific diversification processes could lead certain clades to be systematically further from demographic equilibrium, and thus have lower diversity than expected for their census population size.

## How could selection still explain Lewontin's Paradox?

Even assuming selection parameters estimated from *Drosophila melanogaster*, where the effects of linked selection are thought to be especially strong, the predicted patterns of diversity under linked selection poorly fit observed patterns of diversity across species. My results support the analysis by *Coop, 2016* showing that levels of $\pi_0$ estimated by *Corbett-Detig et al., 2015* are not decoupled from genome-wide average $\pi$, as would occur if linked selection were to explain Lewontin's Paradox. Additionally, my analysis goes a step further, showing that current linked selection models under a wide range of selection parameters are incapable of explaining the observed relationship between census size and diversity. This is in part because mid-$N_c$ species have sufficiently long recombination map lengths to diminish the effects of even strong selection. Overall, while this suggests hitchhiking and background selection seem unlikely to explain patterns of diversity across taxa, there are three major potential limitations of my approach that need further evaluation.

First, I approximate the reduction in diversity using homogeneous background selection and recurrent hitchhiking models (*Kaplan et al., 1989*; *Hudson and Kaplan, 1995*; *Coop and Ralph, 2012*), when in reality, there is genome-wide heterogeneity in functional density, recombination rates, and the adaptive substitutions across species. Each of these factors mediate how strongly linked selection impacts diversity across the genome. Despite these model simplifications, the predicted reduction in diversity in *Drosophila melanogaster* is 85% (when using $N_e$, not $N_c$), which is reasonably close to the estimated 77% from the more realistic model of Elyashiv et al. that accounts for the actual position of substitutions, annotation features, and recombination rate heterogeneity (though it should be noted that these both use the same parameter estimates). Furthermore, even though my model fails to capture the heterogeneity of functionality density and recombination rate in real genomes, it is still conservative, likely overestimating the effects of linked selection to see if it could be capable of decoupling diversity from census size and explain Lewontin's Paradox. This is in part because the strong selection parameter estimates from *Drosophila melanogaster* used, but also because I assume that the effective population size is equal to the census size. Even then, this decoupling only occurs in very high–census-size species, and implies that the diversity in the absence of linked selection, $\pi_0$, is currently underestimated by several orders of magnitude. Moreover, the study of *Corbett-Detig et al., 2015* did consider recombination rate and functional density heterogeneity in estimating the reduction due to linked selection across species, yet their predicted reductions are orders of magnitude weaker than those considered here by assuming that $N_e = N_c$ (*Figure 4—figure supplement 4B*). Overall, given the effects estimated under more realistic inference models are still orders of magnitude weaker than those used in this study, current models of linked selection seem fundamentally unable to fit the diversity–census-size relationship.

Second, my model here only considers hard sweeps, and ignores the contribution of soft sweeps (e.g. from standing variation or recurrent mutations; *Hermisson and Pennings, 2005*; *Pennings and Hermisson, 2006*), partial sweeps (e.g those that do not reach fixation), and the interaction of sweeps and spatial processes. While future work exploring these alternative types of sweeps is needed, the predicted reductions in diversity found here under the simplified sweep model are likely relatively robust to these other modes of sweeps for a few reasons. First, the shape of the diversity–recombination curve is equivalent under models of partial sweeps and hard sweeps, though these imply different rates of sweeps (*Coop and Ralph, 2012*). Second, in the limit where most fitness variation is due to weak soft sweeps from standing variation scattered across the genome (i.e. due to polygenic fitness variation), levels of diversity are well approximated by quantitative genetic linked selection models (*Robertson, 1961*; *Santiago and Caballero, 1995*). The reduction in diversity under these models is nearly identical to that under background selection models, in part because deleterious alleles at mutation-selection balance constitute a considerable component of fitness variation (see Appendix Section B; *Charlesworth and Hughes, 2000*; *Charlesworth, 2015*). Third, the parameters from *Elyashiv et al., 2016* could reflect a mixture of types of sweeps (*Elyashiv et al., 2016* p. 14 and p. 19 of their Supplementary Online Materials). Finally, I also disregarded the interaction of sweeps and spatial processes. For populations spread over wide ranges, limited dispersal slows the spread of sweeps, allowing for new beneficial alleles to arise, spread, and compete against other segregating beneficial variants (*Ralph and Coop, 2015*; *Ralph and Coop, 2010*). Through limited dispersal should act to ''soften sweeps' and not impact my findings for the reasons described

above, future work could investigate how these processes impact diversity in ways not captured by hard sweep models.

Third, other selective processes, such as fluctuating selection or hard selective events (i.e. selection resulting in a reduction in the population size), could reduce diversity in ways not captured by the background selection and hitchhiking models. Since frequency-independent fluctuating selection reduces diversity under most conditions (*Novak and Barton, 2017*), this could lead seasonality and other sources of temporal heterogeneity to reduce diversity in large-$N_c$ species with short generation times more than longer-lived species with smaller population sizes. Future work could consider the impact of fluctuating selection on diversity under simple models (*Barton, 2000*) if estimates of key parameters governing the rate of such fluctuations were known across taxa. Additionally, another mode of selection that could severely reduce diversity across taxa, yet remains unaccounted for in this study, is periodic hard selective events. These selective events could occur regularly in a species' history yet be indistinguishable from demographic bottlenecks with just population genomic data.

## Spatial and demographic processes

One limitation of this study is the inability to quantify the impact of spatial and demographic population genetic processes on the relationship between diversity and census population sizes across taxa. The genomic diversity estimates collated in this study unfortunately lack details about the sampling process and spatial data, which can have a profound impact on population genomic summary statistics (*Battey et al., 2020*). These issues could systematically bias species-wide diversity estimates; for example, if diversity estimates from a cosmopolitan species were primarily from a single region or subpopulation, diversity would be an underestimate relative to the entire population. However, biased spatial sampling alone seems incapable of explaining the $\pi$-$N_c$ divergence in high-$N_c$ taxa. In the extreme scenario in which only one subpopulation was sampled, $F_{ST}$ would need to be close to one for population subdivision alone to sufficiently reduce the total population heterozygosity to explain the orders-of-magnitude shortfall between predicted and observed diversity levels. This can be seen by rearranging the expression for $F_{ST}$ as $H_S = (1 - F_{ST})H_T$, where $H_S$ and $H_T$ are the subpopulation and total population heterozygosities; if $H_T = 4N_c\mu$, then only $F_{ST} \approx 1$ can reduce $H_S$ several orders of magnitude. Yet, across-taxa surveys indicate that $F_{ST}$ is almost never this high within species (*Roux et al., 2016*). Future work could quantify the extent to which more realistic spatial processes contribute to Lewontin's Paradox. For example, high-$N_c$ taxa usually experience range expansions, with repeated founder effects and local extinction/recolonization dynamics that depress diversity (*Slatkin, 1977*). In particular, with the appropriate data, one could estimate the empirical relationship between dispersal distance, range size, and coalescent effective population size across taxa.

In this study, I have focused entirely on assessing the role of linked selection, rather than demography, in reducing diversity across taxa. In contrast to demographic models, models of linked selection have comparatively fewer parameters and more readily permit rough estimates of diversity reductions across taxa. Given that I find that models of linked selection are incapable of explaining the observed relationship between $N_c$ and $\pi$, this supports the hypothesis the diversity across species are shaped primarily by past demographic fluctuations. Still, a full resolution of Lewontin's Paradox would require understanding how the demographic processes across taxa with incredibly heterogeneous ecologies and life histories transform $N_c$ into $N_e$. With population genomic data becoming available for more species, this could involve systematically inferring the demographic histories of tens of species and looking for correlations in the frequency and size of bottlenecks with $N_c$ across species.

## Measures of effective population size, Timescales, and Lewontin's Paradox

Lewontin's Paradox describes the extent to which the effective population sizes implied by diversity, $\widetilde{N}_e$, diverge from census population sizes. However, there are a variety other effective population size estimators calculable from different data and summary statistics (*Wang et al., 2016*; *Caballero, 1994*; *Galtier and Rousselle, 2020*). These include estimators based on the site frequency spectrum, observed decay in linkage disequilibrium, or temporal estimators that use the variance in allele frequency change through time. These various estimators capture different summaries of

effective population size on shorter timescales than coalescent-based estimators (see *Wang, 2005* for a review), and thus could be used to tease apart processes that impact the $N_e$-$N_c$ relationship in the more recent past.

Temporal $N_e$ estimators already play an important role in understanding another summary of the $N_e$-$N_c$ relationship: the ratio $N_e/N_c$, which is an important quantity in conservation genetics (*Frankham, 1995*; *Mace and Lande, 1991*) and in understanding evolution in highly fecund marine species. Surveys of the short-term $N_e/N_c$ relationship across taxa indicate mean $N_e/N_c$ is on order of $\approx 0.1$ (*Frankham, 1995*; *Palstra and Ruzzante, 2008*; *Palstra and Fraser, 2012*), though the uncertainty in these estimates is high, and some species with sweepstakes reproduction systems like Pacific Oyster (*Crassostrea gigas*) can have $N_e/N_c \approx 10^{-6}$ (*Hedgecock, 1994*). Estimates of the $N_e/N_c$ ratio may be an important, yet under appreciated piece of solving Lewontin's Paradox. For example, if $N_e$ is estimated from the allele frequency change across a single generation (i.e. *Waples, 1989*), $N_e/N_c$ constrains estimates of the variance in reproductive success (*Wright, 1938*; *Nunney, 1993*; *Nunney, 1996*). This implies that apart from species with sweepstakes reproductive systems, the variance in reproductive success each generation (whether heritable or non-heritable) is likely insufficient to significantly contribute to constraining $\widetilde{N}_e$ for most taxa. Still, further work is needed to characterize (1) how $N_e/N_c$ varies with $N_c$ across taxa (though see *Palstra and Fraser, 2012*, **Figure 2**), and (2) the variance of $N_e/N_c$ over longer time spans (i.e. how periodic sweepstakes reproductive events act to constrain $N_e$). Overall, characterizing how $N_e/N_c$ varies across taxa and correlates with ecology and life history traits could provide clues into the mechanisms that leads propagule size and survivorship curves to be predictive of diversity levels across taxa (*Romiguier et al., 2014*; *Hallatschek, 2018*; *Barry et al., 2020*).

Finally, short-term temporal $N_e$ estimators may play an important role in resolving Lewontin's Paradox. These estimators, along with short-term estimates of the impact of linked selection (*Buffalo and Coop, 2019*; *Buffalo and Coop, 2020*), can inform us how much diversity is depressed by selection on shorter timescales, free from the rare strong selective events or severe bottlenecks that impact pairwise diversity. It could be that in any one generation, selection contributes more to the variance of allele frequency changes than drift, yet across-taxa patterns in diversity are better explained processes acting sporadically on longer timescales, such as colonization, founder effects, and bottlenecks. Thus, the pairwise diversity may not give us the best picture of the generation to generation evolutionary processes acting in a population to change allele frequencies. Furthermore, certain observed adaptations occur at a pace that is inexplicable given small effective population sizes implied by diversity, and are only possible if short-term effective population sizes are orders of magnitude larger (*Karasov et al., 2010*; *Barton, 2010*).

## Conclusions

In *Building a Science of Population Biology* (*Lewontin et al., 2004*), Lewontin laments the difficulty of uniting population genetics and population ecology into a cohesive discipline of population biology. Lewontin's Paradox of Variation remains a major unsolved problem at the nexus of these two different disciplines: we fail to understand the processes that connect a central parameter of population ecology, census size, to a central parameter of population genetics, effective population size across species. Given that selection seems to fall short in resolving Lewontin's Paradox, a full resolution will require a mechanistic understanding the ecological, life history, and macroevolutionary processes that connect $N_c$ to $N_e$ across taxa. While I have focused exclusively on metazoan taxa since their population densities are more readily approximated from body mass, a full resolution must also include plant species (with the added difficulties of variation in selfing rates, different dispersal strategies, pollination, etc.).

Looking at Lewontin's Paradox through an macroecological and macroevolutionary lens begets interesting questions outside of the traditional realm of population genetics. Here, I have found that diversity and $N_c$ have a consistent relationship without many outliers, despite the wildly disparate ecologies, life histories, and evolutionary histories of the taxa included. Furthermore, taxa with very large census sizes have surprisingly low diversity. Is this explained by macroevolutionary processes, such as different rates of speciation for large-$N_c$ taxa? Or, are the levels of diversity we observe today an artifact of our timing relative to the last glacial maximum, or the last major extinction? Did large-$N_c$ prehistoric animal populations living in other geological eras have higher levels of diversity

than our present taxa? Or, does ecological competition occur on shorter timescales such that strong population size contractions transpire and depress diversity, even if a species is undisturbed by climatic shifts or mass extinctions? Overall, patterns of diversity across taxa are determined by many overlaid evolutionary and ecological processes occurring on vastly different timescales. Lewontin's Paradox of Variation may persist unresolved for some time because the explanation requires synthesis and model building at the intersection of all these disciplines.

## Materials and methods

### Diversity and map length data

The data used in this study are collated from a variety of previously published surveys. Of the 172 taxa with diversity estimates, 14 are from *Corbett-Detig et al., 2015*, 96 are from *Leffler et al., 2012*, and 62 are from *Romiguier et al., 2014*. The Corbett-Detig et al. data is estimated from fourfold degenerate sites, the Romiguier et al. data is synonymous sites, and the Leffler et al. data is estimated predominantly from silent, intronic, and non-coding sites. All types of diversity estimates from *Leffler et al., 2012* were included to maximize the taxa in the study, since the variability of diversity across functional categories is much less than the diversity across taxa. Multiple diversity estimates per taxa were averaged. The total recombination map length data were from both (*Stapley et al., 2017*; 127 taxa), and (*Corbett-Detig et al., 2015*; 9 taxa). Both studies used sex-averaged recombination maps estimated with cross-based approaches; in some cases errors in the original data were found, documented, and corrected. These studies also included genome size estimates used to create *Figure 4—figure supplement 2* and *Figure 4—figure supplement 1*.

### Macroecological estimates of population size

A rough approximation for total population size (census size) is $N_c = DR$, where $D$ is the population density in individuals per km$^2$ and $R$ is the range size in km$^2$. Since population density estimates are not available for many taxa included in this study, I used the macroecological abundance-body size relationship to predict population density from body size. Since body length measurements are more readily available than body mass, I collated body length data from various sources (see https://github.com/vsbuffalo/paradox_variation; copy archived at swh:1:rev:8fa6b5834f6536319-b1e5cd9722ca02d317183df, *Buffalo, 2021*); body lengths were averaged across sexes for sexually dimorphic species, and if only a range of lengths was available, the midpoint was used.

Then, I re-estimated the relationship between body mass and population density using the data in the appendix table of *Damuth, 1987*, which includes 696 taxa with body mass and population density measurements across mammals, fish, reptiles, amphibians, aquatic invertebrates, and terrestrial arthropods. Though the abundance-body size relationship can be noisy at small spatial or phylogenetic scales (Chapter 5, *Gaston and Blackburn, 2008*), across deeply diverged taxa such as those included in this study and *Damuth, 1987*, the relationship is linear and homoscedastic (see *Figure 1—figure supplement 1*). Using Stan (*Stan Development Team, 2020*), I jointly estimated the relationship between body mass from body length using the *Romiguier et al., 2014* taxa, and used this relationship to predict body mass for the taxa in this study. These body masses were then used to predict population density simultaneously, using the *Damuth, 1981* relationship. The code of this routine (pred_popsize_missing_centered.stan) is available in the GitHub repository (https://github.com/vsbuffalo/paradox_variation/).

To estimate range, I first downloaded occurrence records from Global Biodiversity Information Facility (*Global Biodiversity Information Facility, 2020*) using the rgbif R package (*Chamberlain et al., 2014*; *Chamberlain and Boettiger, 2017*). Using the occurrence locations, I inferred whether a species was marine or terrestrial, based on whether the majority of their recorded occurrences overlapped a continent using rnaturalearth and the sf packages (*South, 2017*; *Pebesma, 2018*). For each taxon, I estimated its range by finding the minimum α-shape containing these occurrences. The α parameters were set more permissive for marine species since occurrence data for marine taxa were sparser. Then, I intersected the inferred ranges for terrestrial taxa with continental polygons, so their ranges did not overrun landmasses (and likewise with marine taxa and oceans). I inspected diagnostic plots for each taxa for quality control (all of these plots are available in paradox_variation GitHub repository), and in some cases, I manually adjusted the α parameter or

manually corrected the range based on known range maps (these changes are documented in the code data/species_ranges.r and data/species_range_fixes.r). The range of *C. elegans* was conservatively approximated as the area of the Western US and Western Europe based on the map in *Frézal and Félix, 2015*. *Drosophila* species ranges are from the *Drosophila* Speciation Patterns website, (*Yukilevich, 2012*; *Yukilevich, 2017*). To further validate these range estimates, I have compared these to the qualitative range descriptions *Leffler et al., 2012* (*Figure 1—figure supplement 4*) and compared my α-shape method to a subset of taxa with range estimates from IUCN Red List (*Chamberlain, 2020*; *IUCN, 2020*; *Figure 1—figure supplement 3*). Each census population size is then estimated as the product of range and density.

### Population size validation

I validated the approximate census sizes by comparing the implied biomass of these estimates to estimates of the total carbon biomass on earth by phylum (*Bar-On et al., 2018*). For species $i$ with wet body mass $m_i$ and census size $N_i$, the implied biomass is $m_i N_i$. For all species in a phylum $S$, this total sample biomass is $b_S = \sum_{i \in S} m_i N_i$. I then compare this wet biomass to the carbon biomasses by phylum by *Bar-On et al., 2018*. Across animal species, the ratio of dry to wet body mass, and carbon body mass dry body mass varies little. In their study, Bar-On et al. assume wet body mass has a 70% water content, and 50% of dry body mass is carbon mass, leading to a wet body mass to carbon mass factor of $1 - 0.7/0.5 = 0.15$. I use this factor to convert the total wet biomass to carbon biomass per phylum.

First, I compared the relative carbon biomass in this study to the relative carbon biomass on earth per phylum. This shows that this study's sample over represents chordate biomass (by a factor of ~3), and under represents in arthropod biomass (by a factor of 0.02) relative to the proportion of carbon biomass of these phyla on earth (see column eight of *Table 1*). Second, to check whether the carbon biomass per phylum in the sample was broadly consistent with the total on earth by phylum ($B_S$ for phylum $S$), I calculated the expected sample biomass if species were sampled randomly from the total species in a phylum, ($B_S \times n_S/T_S$, where $n_S$ is the total number of species in the sample in phylum $S$, $T_S$ is the total number of species in phylum $S$ on earth). The fraction of total species on earth included in the sample in this study is depicted in *Figure 1—figure supplement 2*.

Next, I look at the ratio of sample biomass per phylum, $b_S$, to this expected biomass per phylum (*Table 1*). The consistency is quite close for this rough approach and the non-random sample of taxa included in this study. The carbon biomass estimates for chordates implied by the census size estimates are ~24-fold higher than expected, but is well within reasonable expectations given that the chordate sample includes many larger-bodied domesticated species (and is a biased sample in other ways). Similarly, the implied arthropod carbon biomass is quite close to what one would expect. Overall, these values indicate that the census size estimates here do not lead to implied biomasses

**Table 1.** How the total carbon biomass estimates by phylum from *Bar-On et al., 2018* compare to the implied biomass estimates from this study.

All biomass estimates are carbon biomass, and the proportions are of total biomass with respect to the study. The proportion of biomass in this study compared to the Bar-On et al. estimates *Bar-On et al., 2018* indicates chordates are overrepresented and arthropods are underrepresented in the present study; the factor that each phylum is overrepresented is given in the eighth column. Total species by phylum estimates are from *Reaka-Kudla et al., 1996*; *Nicol, 1969*; *Zhang, 2013*; *Chapman, 2009*. The ratio column is the ratio of total biomass implied by the $N_c$ estimates of each species in a phylum to the actual biomass of that phylum.

| phylum | | Bar-On et al. | | Present study | | | | | |
| | total species (T) | biomass (B) | prop. biomass | biomass (b) | prop. biomass | num. species (n) | factor overrepresented | prop. total species (f=n/T) | factor (b/fB) |
| --- | --- | --- | --- | --- | --- | --- | --- | --- | --- |
| Arthropoda | $1.26 \times 10^6$ | 1.20 | 0.4635 | $2.80 \times 10^{-4}$ | 0.0102 | 68 | 0.02 | $5.41 \times 10^{-5}$ | 4.31 |
| Chordata | $5.41 \times 10^4$ | 0.87 | 0.3357 | $2.67 \times 10^{-2}$ | 0.9715 | 68 | 2.89 | $1.26 \times 10^{-3}$ | 24.40 |
| Annelida | $1.70 \times 10^4$ | 0.20 | 0.0772 | $1.23 \times 10^{-5}$ | 0.0004 | 3 | 0.01 | $1.76 \times 10^{-4}$ | 0.35 |
| Mollusca | $9.54 \times 10^4$ | 0.20 | 0.0772 | $4.56 \times 10^{-4}$ | 0.0166 | 13 | 0.21 | $1.36 \times 10^{-4}$ | 16.70 |
| Cnidaria | $1.60 \times 10^4$ | 0.10 | 0.0386 | $3.07 \times 10^{-5}$ | 0.0011 | 2 | 0.03 | $1.25 \times 10^{-4}$ | 2.45 |
| Nematoda | $2.50 \times 10^4$ | 0.02 | 0.0077 | $4.03 \times 10^{-6}$ | 0.0001 | 1 | 0.02 | $4.00 \times 10^{-5}$ | 5.03 |

per phylum that are outside the range of plausibility. For other population size consistency checks, see Appendix 3.

## Phylogenetic comparative methods

Of the full dataset of 172 taxa with diversity and population size estimates, a synthetic calibrated phylogeny was created for 166 species that appear in phylogenies in DateLife project (*O'Meara et al., 2020*; *Sanchez-Reyes and O'Meara, 2019*). This calibrated synthetic phylogeny was then subset for the analyses based on what species had complete trait data. The diversity-population size relationship assessed by a linear phylogenetic mixed-effects model implemented in Stan (*Stan Development Team, 2020*), according to the methods described in *de Villemereuil and Nakagawa, 2014*, (see stan/phylo_mm_regression.stan in the GitHub repository). This same Stan model was used to estimate the same relationship between arthropod, chordate, and mollusc subsets of the data, though a reduced model was used for the chordate subset due to identifiability issues leading to poor MCMC convergence (*Figure 3—figure supplement 1*).

The relationship between recombination map length and the logarithm of population size is non-linear and heteroscedastic, and was fit using a lognormal phylogenetic mixed-effects model on the 130 species with complete data. Since social insects have longer recombination map lengths (*Wilfert et al., 2007*), social taxa were excluded when fitting this model. All Rhat (*Vehtari et al., 2019*) values were below 1.01 and the effective number of samples was over 1,000, consistent with good mixing; details about the model are available in the GitHub repository (phylo_mm_lognormal. stan). Continuous trait maps (*Figure 3A*, *Figure 3—figure supplement 3*, and *Figure 3—figure supplement 2*) were created using phytools (*Revell, 2012*). Node-height tests were implemented based on the methods in Geiger (*Pennell et al., 2014*; *Harmon et al., 2008*), and use robust regression to fit a linear relationship between phylogenetic independent contrasts and branching times.

## Predicted reductions in diversity

The predicted reductions in diversity due to linked selection are approximated using selection and deleterious mutation parameters from *Drosophila melanogaster*, and the recombination map length estimates from *Stapley et al., 2017* and *Corbett-Detig et al., 2015*. The mathematical details of the simplified sweep model are explained in the Appendix Section A. I use estimates of the number of substitutions, $m$, in genic regions between *D. melanogaster* and *D. simulans* from *Hu et al., 2013*. Following *Elyashiv et al., 2016*, only substitutions in UTRs and exons are included, since they found no evidence of sweeps in introns. Then, I average over annotation classes to estimate the mean proportion of substitutions that are beneficial, $\alpha_{\mathrm{Dmel}} = 0.42$, which are consistent with the estimates of Elyashiv et al. and estimates from MacDonald–Kreitman test approaches (see *Eyre-Walker, 2006*, *Table 1*). Then, I use divergence time estimates between *D. melanogaster* and *D. simulans* of $4.2 \times 10^6$ and estimate of ten generations per year (*Obbard et al., 2012*), calculating there are $\gamma_{\mathrm{Dmel}} = \alpha m/2T = 2.26 \times 10^{-3}$ substitutions per generation. Given the length of the *Drosophila* autosomes, $G$, this implies that the rate of beneficial substitutions per basepair, per generation is $\nu_{BP,Dmel} = \gamma_{\mathrm{Dmel}}/G = 2.34 \times 10^{-11}$. Finally, I estimate $J_{\mathrm{Dmel}} \approx 4.5 \times 10^{-4}$ from the estimate of genome-wide average rate of sweeps from Elyashiv et al. (Supplementary Table S6) and assuming *Drosophila* $N_e = 10^6$. These *Drosophila melanogaster* hitchhiking parameter estimates are close to other previously-published estimates (*Figure 4—figure supplement 5*). Finally, I use $U_{\mathrm{Dmel}} = 1.6$, from *Elyashiv et al., 2016*. With these parameter estimates from *D. melanogaster*, the recombination map lengths across species, and *Equation (1)*, I estimate $\pi_{\mathrm{BGS+HH}}$ (assuming $N_c = N_c$) across all species. This leads to a range of predicted diversity ranges across species corresponding to $\mu = 10^{-9}$–$10^{-8}$; to visualize these, I take a convex hull of all diversity ranges and smooth this with R's smooth. spline function.

## Acknowledgements

I would like to thank Andy Kern and Peter Ralph for helpful discussions and supporting me during this work, and Graham Coop for inspiration and helpful feedback during socially distanced nature walks at Yolo Basin. I thank Jessica Stapley for kindly providing the recombination map length data, and Yaniv Brandvain, Amy Collins, Doc Edge, Tyler Kent, Chuck Langley, Matt Osmond, Sally Otto,

Molly Przeworski, Jeff Ross-Ibarra, Aaron Stern, Anastasia Teterina, Michael Turelli, Margot Wood, and my Kern-Ralph labmates for helpful discussions. Sarah Friedman, Katherine Corn, and Josef Uyeda provided very useful advice about phylogenetic comparative methods; yet I take full responsibility for any shortcomings of my analysis. Finally, I am indebted to Guy Sella, Matt Pennell, and two other anonymous reviewers for helpful feedback. I would like to also thank UO librarian Dean Walton for helping me track down some rather difficult to find older papers. This work was supported by an NIH Grant (1R01GM117241) awarded to Andrew Kern.

## Additional information

### Funding

| Funder | Grant reference number | Author |
|---|---|---|
| National Institutes of Health | 1R01GM117241 | Vince Buffalo |

The funders had no role in study design, data collection and interpretation, or the decision to submit the work for publication.

### Author contributions
Vince Buffalo, Conceptualization, Resources, Data curation, Software, Formal analysis, Validation, Investigation, Visualization, Methodology, Writing - original draft, Writing - review and editing

### Author ORCIDs
Vince Buffalo https://orcid.org/0000-0003-4510-1609

### Decision letter and Author response
Decision letter https://doi.org/10.7554/eLife.67509.sa1
Author response https://doi.org/10.7554/eLife.67509.sa2

## Additional files
### Supplementary files
• Transparent reporting form

### Data availability
All primary datasets collated by this study, including new census size and range estimates, are available on Github at http://github.com/vsbuffalo/paradox_variation (copy archived at https://archive.softwareheritage.org/swh:1:rev:8fa6b5834f6536319b1e5cd9722ca02d317183df). An archived version of this repository is also available at Zenodo.

The following dataset was generated:

| Author(s) | Year | Dataset title | Dataset URL | Database and Identifier |
|---|---|---|---|---|
| Vince B | 2021 | vsbuffalo/paradox_variation: biorxiv v.1 with minor corrections | https://doi.org/10.5281/zenodo.4542480 | Zenodo, 10.5281/zenodo.4542480 |

The following previously published datasets were used:

| Author(s) | Year | Dataset title | Dataset URL | Database and Identifier |
|---|---|---|---|---|
| Stapley J, Feulner PGD, Johnston SE, Santure AW, Smadja CM | 2017 | Supplementary material from "Variation in recombination frequency and distribution across eukaryotes: patterns and processes" | https://doi.org/10.6084/m9.figshare.c.3904942.v3 | figshare, 10.6084/m9.figshare.c.3904942.v3 |

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

## Appendix 1

### Simplified sweep effects model

I use a simplified model of the effects of recurrent hitchhiking and background selection (BGS) occurring uniformly along a genome. Expected diversity is given by

$$E(\pi) = \frac{\theta}{\theta + 1/B + 2NS} \tag{2}$$

$$\approx \frac{\theta}{1/B + 2NS} \tag{3}$$

(Equation 1 *Elyashiv et al., 2016*, Equation 4 of *Kim and Stephan, 2000*, and Equation 20 of *Coop and Ralph, 2012*). The BGS component is given by *Hudson and Kaplan, 1995*,

$$B(U,L) = N_e \exp\left(-\frac{U}{L}\right) \tag{4}$$

and the hitchhiking component is

$$S = \frac{\nu_{\mathrm{BP}}}{r_{\mathrm{BP}}} J \tag{5}$$

(*Coop and Ralph, 2012*, Equation 20) where $\nu_{\mathrm{BP}}$ and $r_{\mathrm{BP}}$ are the substitutions and recombination per basepair respectively, $J$ is the probability that two lineages coalesce down to one, given sweeps occur uniformly along the genome. Under this homogeneous sweep model, $J$ is

$$J = \int_0^L q_f(r)^2 dr \tag{6}$$

where $q_f(r)$ is the approximate probability that a lineage is trapped by a sweep to frequency $f$ when it is $r$ recombination fraction away from this sweep (*Coop and Ralph, 2012*; Equation 15).

Since I use *Drosophila melanogaster* parameter estimates from *Elyashiv et al., 2016*, I now reconcile their model's $S$ term with the simple model above. They estimate $S$ in *Drosophila melanogaster* using a composite likelihood model that considers hitchhiking and background selection simultaneously, using substitutions and stratifying by annotation. For a neutral position at site $x$, the coalescence rate due to sweeps is given by Elyashiv et al.'s Equation 3,

$$S(x) = \frac{1}{T} \sum_{i_S} \alpha(i_S) \sum_{y \in a(i_S)} \int \exp(-r(x,y)\tau(s,N)) g(s|i_S) ds \tag{7}$$

where $T$ is the length of the lineage (in generations) on which substitutions accrue, $i_S = 1, \ldots, I_S$ is the annotation class (e.g. exons, introns, UTRs), $\alpha(i_S)$ is the fraction of substitutions in annotation class $i_S$ that are beneficial, $a(i_S)$ is the set of all substitutions in annotation class $i_S$, $\tau(s,N)$ is the fixation time of a site with additive effect $s$, and $g(s|i_S)$ is the distribution of selection coefficients for annotation class $i_S$.

Note, that we can recover the model of *Coop and Ralph, 2012* from this expression. Suppose there is only one annotation class, and $\alpha$ fraction of substitutions are beneficial, and one selection coefficient $\bar{s}$, (i.e. $g(s) = \delta_0(s - \bar{s})$), then

$$S(x) = \frac{\alpha}{T} \sum_{y \in a} \exp(-r(x,y)\tau(\bar{s},N)). \tag{8}$$

Let the number of substitutions be $m := |a|$, and imagine their positions are uniformly distributed on a segment of length $G$ basepairs with the focal site is the middle at position $x = 0$. Then, each substitution $y$ is a random distance $l_y \sim U(-G/2, G/2)$ away from the focal site. Assuming the recombination rate is a constant $r_{\mathrm{BP}}$ per basepair, and approximating the sum with an integral, we have,

$$S = \frac{\alpha}{T} \sum_{i=1}^{m} E_{l_i}(\exp(-r_{\mathrm{BP}} l_i \tau(\bar{s}, N))) \tag{9}$$

$$= \frac{\alpha}{TG} \sum_{i=1}^{m} \int_0^G \exp(-r_{\mathrm{BP}} \ell \tau(\bar{s}, N)) d\ell \tag{10}$$

$$= \frac{\alpha m}{TG} \int_0^G \exp(-r_{\mathrm{BP}} \ell \tau(\bar{s}, N)) d\ell \tag{11}$$

Using $u$-substitution with $r = \ell r_{\mathrm{BP}}$ this simplifies to

$$S = \frac{\alpha m}{TGr_{\mathrm{BP}}} \int_0^L \exp(-r \tau(\bar{s}, N)) dr \tag{12}$$

where $L = Gr_{\mathrm{BP}}$.

To simplify this notation, note that the rate of adaptive substitutions per basepair per generation is $\nu_{\mathrm{BP}} = \alpha m / GT$, so

$$S = \frac{\nu_{\mathrm{BP}}}{r_{\mathrm{BP}}} \int_0^L \exp(-r \tau(\bar{s}, N)) dr \tag{13}$$

This is analogous to the second term of *Coop and Ralph, 2012*, Equation 17, with $k = i = 2$ and $x = 1$ (e.g. conditioning on a sweep to fixation). Note that there appears to be a factor of two error in *Elyashiv et al., 2016* compared to *Coop and Ralph, 2012*; here I include the factor of two. Then,

$$S = \frac{\nu_{\mathrm{BP}}}{r_{\mathrm{BP}}} \underbrace{\int_0^L \exp(-2r \tau(\bar{s}, N)) dr}_{J} \tag{14}$$

where the integral is equal to $J$ ($J_{2,2}$ of Equation 15 in *Coop and Ralph, 2012*) since a simple model of $q_f(r) = f \exp(-2r \tau(s, N))$ and if we condition on fixation, $f = 1$. This expression is useful to generalize across species, since we know $N$ and $L$. Additionally, we have estimates of $\alpha$ and $m/T$ in *Drosophila* and other species. In Elyashiv et al, they consider the number of substitutions per generation in genic regions only; it should be noted that the number of coding basepairs varies little across species. For convenience, I define $\gamma = \alpha m / T$ as the number of adaptive substitutions per generation per entire genome, such that $S(\gamma, J, L) = \frac{\gamma}{L} J$ used in the main text. Using the estimates of $m \approx 4.5 \times 10^5$, $\alpha \approx 0.42$, and $T \approx 8.4 \times 10^7$ from the Supplementary Material of Elyashiv et al., I arrive at $\gamma \approx 0.00226$ adaptive substitutions per generation, per genome. For a $\approx 100$ megabase genome, this translates to a $\nu_{\mathrm{BP}} \approx 2.34 \times 10^{-11}$, which is close to previous estimates (*Figure 4—figure supplement 5A*). For $J$, I use an empirical estimate calculated from the genome-wide average of the rate of coalescent events due to sweeps, from Supplementary Table S6 of Elyashiv et al. ($r_s = 2NS \approx 0.92$; see *Figure 4—figure supplement 5B*). This implies $J \approx 4.46 \times 10^{-4}$. Alternatively, I have tried using the estimated distribution of selection coefficients from Elyashiv et al., but this led to a weaker estimate of $J$, since the adaptive substitutions considered tend to cluster around genic regions.

## Appendix 2

### Background selection and polygenic fitness models

Throughout the main text, I use recurrent hitchhiking and background selection models to estimate the reduction in diversity due to linked selection. Another class of linked selection models, which I refer to as quantitative genetic linked selection models (QGLS; *Robertson, 1961*; *Santiago and Caballero, 1995*), can also depress genome-wide diversity. Furthermore, these models may depress diversity at neutral sites unlinked to the regions containing fitness variation. While I did not explicitly incorporate these models into my estimates of the diversity reductions, their effect is implicit in background selection models because they are analytically nearly identical. Here, I briefly sketch out the connection between BGS and QGLS models.

Under the *Santiago and Caballero, 1998* model, the effective population size is $N_e^{\text{SC98}} = N \exp(-C^2/(1-Z)L)$, where $C^2$ is the standardized heritable fitness variation, $1-Z$ is the decay of genetic variance through time, and $L$ is the recombination map length. This model can accommodate a variety of modes of selection such as selection on an infinitesimal trait (*Santiago and Caballero, 1995*, p. 1016), and the flux of either weakly advantageous or deleterious alleles (*Santiago and Caballero, 1998*, p. 2109). If the source of fitness variation is entirely the input of new deleterious mutations with heterozygous effect $sh$ at rate $U$ per diploid genome per generation, then under mutation-selection balance, the equilibrium relative variance in reproductive success $C^2 = Ush$ (*Crow and Kimura, 1970*; *Caballero, 2020*, p. 167), and $Z = 1 - sh - 1/2N_c$ (*Santiago and Caballero, 1998*). Thus, if $1/2N_c << sh << 1$, then $C^2/(1-Z) \approx U$ and $N_e^{\text{SC98}} \approx N \exp(-U/L)$, which is the BGS model used in the main text and is a result of many background selection models with similar assumptions (*Hudson and Kaplan, 1994*, Equation 15; *Hudson and Kaplan, 1995*, Equation 9; *Nordborg et al., 1996*, Equation 4; *Barton, 1995*, Equation 22b). Intuitively, the similarity of these models reflects the fact that a substantial proportion of heritable fitness variation is caused by the continual flux of deleterious alleles across the genome under mutation-selection balance (*Charlesworth, 2015*; *Charlesworth and Hughes, 2000*).

## Appendix 3

### Additional population size validation

In addition to the biomass-based validation described in the main text, I also conducted a few other consistency checks. First, note that the body-mass-based estimates of density for *Drosophila* are similar to previously used estimates in surveys of census size and diversity. **Nei and Graur, 1984** suggested a maximum of 5 *Drosophila* per m$^2$, including regions of the range that are not inhabitable. Across *Drosophila*, the body mass based estimates suggest $10^{6.7}$–$10^{7.6}$ individuals per km$^2$, or 4.5 – 36.3 individuals per m$^2$, which are consistent with this previous estimate. Nei and Graur's estimates of *Drosophila pseudoobscura*'s census size are four orders of magnitude smaller than mine, but their approach uses a speculated ratio of population sizes of different *Drosophila* species rather than range sizes (**Nei and Graur, 1984**, p. 81).

As another consistency check, I looked at the rank order of mammals by biomass. Whale species have the first and third highest biomass with 11.4 and 3.9 megatons of carbon biomass (for *Balaenoptera bonaerensis* and *Eschrichtius robustus*, respectively). While this seems high, a recent study shows that across whale species, pre-whaling carbon biomass was at the tens of megatons level (**Pershing et al., 2010**, *Table 1* and *Figure 1*). Given that my census size estimates represent populations at a macroecological equilibrium, they would not reflect reduced density due to whaling or other anthropogenic causes. Humans had the second largest biomass, followed by wolf species (*Canis lupus* and *C. latrans*); as with whales, the population sizes for wolf species represent pre-anthropogenic densities and are overestimates compared to current population sizes, as expected.

Finally, there are other estimates of approximate population sizes for some species that I compared my estimates to. The United Nation's FAOSTAT database estimates the total number of horses (*Equus caballus*) on earth as ~60 million; the estimate in this study is close to 40 million. For other domesticated species like chicken (*Gallus gallus*), estimates range from 25 million to 19.6 billion (**FAOSTAT statistics database, 2021**; **Robinson et al., 2014**); the present study's estimate lies in the middle at ~175 million. Again, this is a known limitation of this method, as the range is estimated from occurrence data and does not consider species' niches. This present study's estimate of the number of king penguins (*Aptenodytes patagonicus*) is about 3 million; the population size was recently estimated as 2.23 million pairs (**Shirihai, 2008**).

## Appendix 4

### Diversity and IUCN Red List Status

I also investigated the relationship between species' IUCN Red List categories (an ordinal scale of how threatened a species is) and both diversity and population size, finding that species categorized as more threatened have both smaller population sizes and reduced diversity, compared to non-threatened species (*Appendix 4—figure 1*) consistent with past work (*Spielman et al., 2004*). A linear model of diversity regressed on population size has lower AIC when the IUCN Red List categories are included, and the estimates of the effect of IUCN status are all negative on diversity, though not all are significant in part because some categories have three or fewer species (*Appendix 4—table 1*).

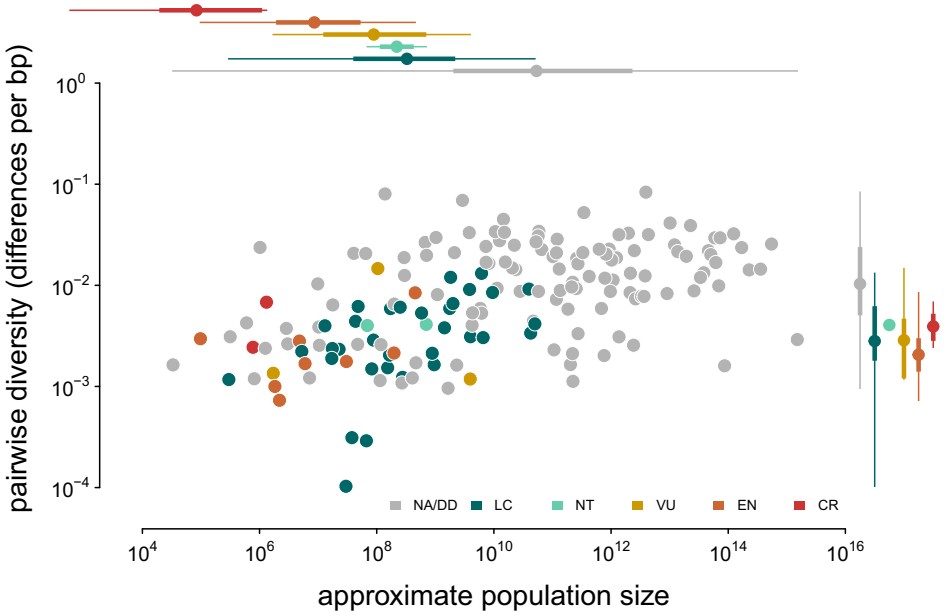

**Appendix 4—figure 1.** A version of *Figure 2* with points colored by their IUCN Red List conservation status. Margin boxplots show the diversity and population size ranges (thin lines) and interquartile ranges (thick lines) for each category. NA/DD indicates no IUCN Red List entry, or Red List status Data Deficient; LC is Least Concern, NT is Near Threatened, VU is Vulnerable, EN is Endangered, and CR is Critically Endangered.

**Appendix 4—table 1.** The regression estimates of full IUCN Red List population size model for diversity, $\log_{10}(\pi) = \beta_0 + \beta_{LC}LC + \beta_{NT}NT + \beta_{VU}VU + \beta_{EN}EN + \beta_{CR}CR + \beta_{N_c}\log_{10}(N_c)$; $df = 165$. Using AIC to compare this full model to a reduced model of $\log_{10}(\pi) = \beta_0 + \beta_{N_c}\log_{10}(N_c)$, $AIC_{\text{full}} = 204.9$, $AIC_{\text{reduced}} = 216.4$.

|  | Mean | 2.5 % | 97.5 % |
| --- | --- | --- | --- |
| $\beta_0$ | −2.80 | −3.20 | −2.50 |
| $\beta_{LC}$ | −0.39 | −0.57 | −0.21 |
| $\beta_{NT}$ | −0.22 | −0.83 | 0.39 |
| $\beta_{VU}$ | −0.34 | −0.84 | 0.16 |
| $\beta_{EN}$ | −0.40 | −0.73 | −0.07 |
| $\beta_{CR}$ | −0.03 | −0.65 | 0.59 |
| $\beta_{N_c}$ | 0.08 | 0.05 | 0.11 |

