## [Decision Letter]

Thank you for submitting your article "Why do species get a thin slice of π? Revisiting Lewontin's Paradox of Variation" for consideration by *eLife*. Your article has been reviewed by 4 peer reviewers, including Guy Sella as the Reviewing Editor and Reviewer #1, and the evaluation has been overseen by Detlef Weigel as the Senior Editor. The following individual involved in review of your submission has agreed to reveal their identity: Matthew Pennell (Reviewer #4).

Essential revisions:

The reviewers appreciated the scholarship and extensive work that went into this manuscript. At the same time, they had several issues with the novel analyses. After some discussion, they suggested that if the paper is revised into more of a review, then it could be of interest to the broad readership of *eLife*. This would imply substantial streamlining, down-weighting and even removing some analyses, and addressing the reviewers comments on others. Specifically:

– Find ways to validate the census size estimates (Reviewers 1 and 2).

– The reviewers had a hard time understanding the motivation for the phylogenetic analysis (e.g., Reviewers 1-3). One option is to clarify this motivation, as well as the interpretation of the results; another is to remove this analysis or parts of it. In addition, Reviewer 4, who is an expert on these analyses, had a number of concrete comments that should be addressed.

– While the reviewers found the analysis of linked selection effects intriguing, they were unclear about its interpretation. Notably, to what extent is it simply affirming the conclusions of Coop (2016), as opposed to illustrating a much more dramatic effect (e.g., Reviewers 1-3).

*Reviewer #1 (Recommendations for the authors):*

1) Substantial streamlining and editing would make the manuscript much clearer.

2) The abstract is way too long: not on the intended resolution.

3) If you aim for a broad readership, you may consider having the background be less of a historical review (without sacrificing scholarship). Also, no need to recap the historical review at the beginning of the Discussion.

4) You recap what you do at length several times (e.g., L 130-156), which is repetitive.

5) It would be clearer to use consistent terminology, e.g., instead of "enigma", "anomaly", "paradox" and "explanation", "resolution"…

6) The writing switches between acknowledging that several factors are plausibly at work and seeking "a solution".

7) It is claimed that "an ordinary least squares relationship on a log-log scale fits the data well" but I did not find a quantification to this effect. Namely, what proportion of the variance does it explain? Moreover, it is claimed that this relationship is homoscedastic. Can that be quantified as well? From looking at the figure it seems that the regression may explain ~1 of the ~3 orders of magnitude in diversity levels and the residuals explain ~2. It would also be helpful to say what we learn from this fit that we did not learn from staring at the plot. Does one (or more) of the potential explanations for Lewontin's paradox posit a log-log relationship?

*Reviewer #2 (Recommendations for the authors):*

I would love to see a revised version of this piece published.

*Reviewer #3 (Recommendations for the authors):*

My main suggestion is to expand a little more the arguments for why the author feels particular factors are unlikely to explain the patterns qualitatively, and to touch on some of the alternative explanations raised in the public review.

[Editors' note: further revisions were suggested prior to acceptance, as described below.]

Thank you for submitting your article "Quantifying Lewontin's Paradox Suggests Natural Selection is Unlikely to Explain the Narrow Range of Diversity Among Species" for consideration by *eLife*. Your article has been reviewed by 4 peer reviewers, including Guy Sella as the Reviewing Editor and Reviewer #1, and the evaluation has been overseen by Detlef Weigel as the Senior Editor. The following individual involved in review of your submission has agreed to reveal their identity: Matthew Pennell (Reviewer #4).

All of us find the topic important, the manuscript very interesting and the revised version improved. At the same time, we also felt that the potential impact of the paper is diminished by not (or only partially) addressing key issues raised in the previous round of review. All the more so, if this is to be more of a research paper than a review (although there can be some range in between).

Notably:

1) We found that the purpose and insights of the phylogenetic regression model were still not clear. While we had Matthew to answer some of our questions, most of the readers will not have such a resource.

2) We think that more precisely articulating how the linked selection analysis may have moved us forward (i.e., beyond just saying that it did not resolve the paradox entirely) would be helpful. For example, if you believe that it accounts for several orders of magnitude even if it does not account for the shape of the dependency of π on Nc for intermediate values of Nc, then that is important progress.

3) In that regard, we also think that exploring a somewhat wider range of selection parameters (along the lines suggested by reviewer 1 in the previous round) would be helpful in elucidating how far linked selection can and cannot take us.

In summary, we think that the impact of your nice work would be enhanced substantially by addressing these issues, but we leave that to your discretion and will not require another round of reviews.

*Reviewer #1 (Recommendations for the authors):*

The manuscript has improved substantially in both substance and presentation. I believe that it would be all the more impactful with another thorough round of editing. I attach a pdf with comments/suggestions, but am not a native speaker and am also mildly dyslexic, which is to say that it may be worth getting additional feedback from better writers/editors.

A few comments about substance:

– The interpretation and necessity of controlling for phylogenetic non-independence. Having read the revised manuscript, your replies and the comments of Reviewer 4, I remain confused about the interpretation of the analysis. For example, I don't understand whether it is about controlling for phylogenetic relationships in errors or in factors that affect the relationship between π and Nc. Moreover, to the extent that it does the latter, what are the assumptions that go into the correction.

In the discussion you say that (l. 415-16) "The evidence of high phylogenetic signal found in this study suggests PCMs are needed, in part to avoid spurious results from phylogenetic pseudo-replication." If I understand you correctly, you are suggesting that factors that modify the relationship between π and Nc, such as life history traits, are likely to be more similar in more closely related species, and consequently, the (pi, Nc) points corresponding to closely related species may be considered replication in that they reflect the same processes. If this is what you meant, I think it should be spelled out. Moreover, it would be good to do so early on, in the results, such that the reader can better understand what the PCM analysis is plausibly about.

In the Results section you say that (l. 220-223): "If the relationship between diversity and population size was free of shared phylogenetic history, \λ = 0 and all the variances could be explained by evolution on the tips; this is analogous to Lynch's conjecture that coalescent times should be free of phylogenetic signal (2011)." Doesn't this contradict the interpretation discussed in the previous paragraph?

Also, I still don't understand the claim that PCM corrects for "spurious pseudo-replication". Shouldn't the determination whether it is "pseudo-replication" depend on the notion of the "true" relationship that you are trying to estimate? Stated differently, maybe some of the phylogenetic signal arises from similarities in factors that affect the pi-Nc relationship and others that just affect π (e.g., mutation rates), a given notion of the "true" relationship would suggest that you want to correct for the latter but not the former. How can PCMs do that without specifying what it is that you are correcting for?

I realize that some of these problems may reflect my ignorance about PCMs, but doubt that the general readership you are aiming is much more knowledgeable.

– Relationship between genetic map length and Nc. You note that (l 333-5): "These findings are consistent with both the hypothesis that non-adaptive processes increase genome size in small-Ne species (Lynch and Conery, 2003) which in turn could increase map lengths…". I think the map length is largely explained (e.g., R^2=0.96) by the requirement of having one cross-over per chromosome or chromosomal arm (can't remember which). Specifically, I think that this relationship is much stronger than with genome size, and am not sure whether there is any residual effect of size after controlling for the number of chromosomes.

– Discussion about different measures of Ne. I found several points in this section deep and insightful. As you point out, there are different N_e's for different quantities, and comparisons among them informs us about different processes. I think it would be helpful to emphasize that these are different quantities rather than estimators of the same quantity and that Lewontin's paradox is specifically about the one defined by diversity levels.

A few general comments about presentation:

– You often use several different terms for the same thing, e.g., N_e and expected coalescence rate; explain, solve and resolve; genetic/recombinational map length, heterozygosity/pairwise diversity. This is confusing to readers, who wonder whether there was a reason for using the specific term. Choosing one for each term and sticking with it would be clearer.

– You sometimes use terms that seem to be private abbreviations, e.g., "strong selection parameters" and "quantifying… paradox".

– Perhaps avoid hyphenations as abbreviations, e.g., "low-Nc species" and "across-taxa relationship".

- The discussion would benefit from extensive editing.

Very nice work!

*Reviewer #2 (Recommendations for the authors):*

I have mixed feelings about this revision. The author did not follow the reviewers' suggestion of shifting from a research to a review article, and rather tried to reinforce his original results. One problem I'm having is that I still do not see the point of the sophisticated phylogenetic analysis that is presented. It is clear from the existing literature that π has a strong phylogenetic inertia; I guess the interesting question is what causes this inertia; whether λ is 0.4, 0.6 or 0.8 is good to know but not a major achievement, in my opinion. I have a similar comment regarding the report that genetic map tends to be shorter in large census sized species, which is cool but not totally novel, and not a very strong effect – I do not understand why social insects are excluded from the model, by the way. I like much figure 4b, which substantiates Coop's 2016 argument via empirical data analysis – but again the added value is mainly in the illustration; the argument existed already.

In my opinion this version has essentially the same strengths and weaknesses as the previous one: text book-like figures and excellent writing, but no breakthrough as far as the newly reported results are concerned. A missed opportunity, I would say.

*Reviewer #3 (Recommendations for the authors):*

In reviewing the author's response letter and the revised draft, the revisions have certainly streamlined the paper and clarified the author's justification for the analyses.

However, I still have some concerns about how the work is motivated at the outset, and how the novelty of the results is explained:

In the Introduction, the bottom of page 3 the author says that a limitation of past work is that others do not propose a mechanism by which traits act to constrain diversity within a few orders of magnitude. This seems to be a major motivation for doing a new study, but how does this study tackle this question? Did the author have an a priori reason to expect that a more explicit estimation of Nc would resolve Lewontin's paradox, and if so, why? Could the phylogenetic correction have been expected to have resolved Lewontin's paradox, and if so, why (see below)?

Correcting for the effect of Nc on map length in tandem with the linked selection model fitting does seem to fall into this category, and with this in mind, I think the author could go a long way towards clarifying the novelty of the work by being more explicit about these results. As noted by reviewer 1 in the first round of reviews the 'glass half full' interpretation of this study for the genetic draft model is that the map length correction combined with linked selection model fitting CAN actually go a significant way towards shrinking down the range of diversity expected. While it can't go ALL the way, this is an important and novel result that is more important to clarify than to simply conclude that we are still in the exact same conclusion zone as before this work. The author has gone some way towards using linked selection to explain the variation, and this is worth clarifying, and perhaps quantifying in the discussion.

I still find the motivation for the phylogenetic correction hard to parse in the introduction, and I would suggest front-loading some of the points the author makes in the response about the importance of phylogenetic correction even if coalescent times themselves are not constrained by phylogeny into the intro. Clearly, multiple reviewers found it difficult to understand why phylogenetic correction was needed and wasn't just eroding power, so this is important to front-load.

Relatedly, I am still not convinced by the argument that for Lynch's conjecture to be true λ must be zero (page 7). A number of possible interpretations made by the author in the results provide possible mechanisms (like phylogenetic changes in mutation rate) that would still enable coalescent times themselves to be free of a direct phylogenetic effect. The revisions and response now provide a better explanation for the importance of a phylogenetic correction in any case, but I don't think any of the analyses have definitively told us that coalescent times have a phylogenetic signal- it could simply be the phylogenetic inertia of traits associated with Ne and mutation rate.

*Reviewer #4 (Recommendations for the authors):*

I liked this paper before and I like it even better now. You have done a thorough and thoughtful job at responding to reviewer comments and have addressed my major critiques. This is a really excellent study.

---

## [Author Response]

Essential revisions:The reviewers appreciated the scholarship and extensive work that went into this manuscript. At the same time, they had several issues with the novel analyses. After some discussion, they suggested that if the paper is revised into more of a review, then it could be of interest to the broad readership of eLife. This would imply substantial streamlining, down-weighting and even removing some analyses, and addressing the reviewers comments on others. Specifically:

Thank you for the feedback. At present (and discussed in more detail in the replies below), the three novel analyses and findings of this study (see reply to Reviewer 2) make this outside the scope of a review, so I am submitting it as a research article.

– Find ways to validate the census size estimates (Reviewers 1 and 2).

Reviewer 1’s feedback helped me discover an error in my previous analysis. The census sizes are now a few orders of magnitude smaller for cosmopolitan arthropod species like *Drosophila melanogaster*. I have also conducted a consistency check by comparing the implied biomasses from my study to previously-published estimates across phyla.

– The reviewers had a hard time understanding the motivation for the phylogenetic analysis (e.g., Reviewers 1-3). One option is to clarify this motivation, as well as the interpretation of the results; another is to remove this analysis or parts of it. In addition, Reviewer 4, who is an expert on these analyses, had a number of concrete comments that should be addressed.

I have significantly reworked how this analysis was framed, in part thanks to the feedback of Reviewer 4. I also have reworked some sections of the discussion to discuss why such analyses are needed.

– While the reviewers found the analysis of linked selection effects intriguing, they were unclear about its interpretation. Notably, to what extent is it simply affirming the conclusions of Coop (2016), as opposed to illustrating a much more dramatic effect (e.g., Reviewers 1-3).

I have reworked part of the discussion (lines 518-524) to discuss how the findings in this paper fits in, and builds off of the findings of Coop (2016). I discuss this in much more detail below.

Reviewer #1 (Recommendations for the authors):1) Substantial streamlining and editing would make the manuscript much clearer.

Thank you, following this and other feedback, I have edited some unclear sections.

2) The abstract is way too long: not on the intended resolution.

I have reduced the abstract significantly, to 200 words (the limit of *eLife*).

3) If you aim for a broad readership, you may consider having the background be less of a historical review (without sacrificing scholarship). Also, no need to recap the historical review at the beginning of the Discussion.

I have removed the recap at the beginning of the discussion. I have not cut down the historical context as, (1) I believe such a mini review (in the spirit of Coop and Ralph, 2012) is needed, and (2) other reviewers and readers did seem to like this aspect.

4) You recap what you do at length several times (e.g., L 130-156), which is repetitive.

I am not sure I follow this comment — this is the first time I describe what I am doing in the manuscript, which seems like an important part of the introduction?

5) It would be clearer to use consistent terminology, e.g., instead of "enigma", "anomaly", "paradox" and "explanation", "resolution"…

"… I have removed some synonyms for clarity.

6) The writing switches between acknowledging that several factors are plausibly at work and seeking "a solution".

I agree that “explanation” or “resolution” are better terms, and have made these changes.

7) It is claimed that "an ordinary least squares relationship on a log-log scale fits the data well" but I did not find a quantification to this effect. Namely, what proportion of the variance does it explain? Moreover, it is claimed that this relationship is homoscedastic. Can that be quantified as well? From looking at the figure it seems that the regression may explain ~1 of the ~3 orders of magnitude in diversity levels and the residuals explain ~2. It would also be helpful to say what we learn from this fit that we did not learn from staring at the plot. Does one (or more) of the potential explanations for Lewontin's paradox posit a log-log relationship?

The log-log relationship is used because of heteroscedasticity on a linear-log scale, and because both axes vary over several orders of magnitude. I do not think it is too surprising that π varies a few orders of magnitude, since Nc varies over so many (and Ne a fraction of that). Regarding the proportion of variance, I have included an R^2^ in the paper (the adjusted R^2^ is about 0.25). However, it should be noted that R^2^ has numerous statistical issues (see p. 181-182 of Cosma Schalizi’s The Truth about Linear Regression, http://www.stat.cmu.edu/~cshalizi/TALR/TALR.pdf). Likewise, I don’t think a formal test of heteroscedasticity is particularly helpful; a plot of the residuals versus x values is convincing enough (though I think unnecessary to include in the supplementary materials).

Reviewer #3 (Recommendations for the authors):My main suggestion is to expand a little more the arguments for why the author feels particular factors are unlikely to explain the patterns qualitatively, and to touch on some of the alternative explanations raised in the public review.

This is useful feedback that other reviewers have voiced as well. I have expanded on how my results connection to that of Coop (2016) on lines 517-524, as well as addressed some more points about the need for PCMs in the discussion (lines 440-473).

[Editors' note: further revisions were suggested prior to acceptance, as described below.]All of us find the topic important, the manuscript very interesting and the revised version improved. At the same time, we also felt that the potential impact of the paper is diminished by not (or only partially) addressing key issues raised in the previous round of review. All the more so, if this is to be more of a research paper than a review (although there can be some range in between).Notably:1) We found that the purpose and insights of the phylogenetic regression model were still not clear. While we had Matthew to answer some of our questions, most of the readers will not have such a resource.

I have significantly edited this (lines 197-284) to more clearly explain the purpose of the phylogenetic mixed-effect model and better frame the statistical issues it addresses.

2) We think that more precisely articulating how the linked selection analysis may have moved us forward (i.e., beyond just saying that it did not resolve the paradox entirely) would be helpful. For example, if you believe that it accounts for several orders of magnitude even if it does not account for the shape of the dependency of π on Nc for intermediate values of Nc, then that is important progress.

The main problem I show here is that assuming strong selection (i.e. the levels predicted under *D. melanogaster*), theoretic linked selection models do not match the observed data if Ne = Nc. Unfortunately, this analysis cannot shed light on how much linked selection impacts diversity across species because I use *Drosophilamelanogaster* parameters that are only suitable for qualitatively checking if linked selection could explain the observed pattern, not quantify the impact across species. Still, my findings are a serious qualitative mismatch between the observed and predicted relationship between diversity and population size. Furthermore, as the new analysis described below shows, this qualitative finding is not sensitive to parameter choice.

3) In that regard, we also think that exploring a somewhat wider range of selection parameters (along the lines suggested by reviewer 1 in the previous round) would be helpful in elucidating how far linked selection can and cannot take us.

I have added a new supplementary figure (p. 47) looking at how a ten-fold increase in BGS (the U parameter) and HH (the γ parameter) would impact the predicted relationship between diversity and population size. This figure also depicts the predicted effect of BGS and HH on selection individually (subfigure A). Overall, this demonstrates that the poor fit found between observed and predicted levels of diversity is not something that can be remedied with stronger parameters. I discuss this on lines 377-386, which also addresses the above issue as well.

In summary, we think that the impact of your nice work would be enhanced substantially by addressing these issues, but we leave that to your discretion and will not require another round of reviews.

Thank you to all the reviewers for feedback.

Reviewer #1 (Recommendations for the authors):The manuscript has improved substantially in both substance and presentation. I believe that it would be all the more impactful with another thorough round of editing. I attach a pdf with comments/suggestions, but am not a native speaker and am also mildly dyslexic, which is to say that it may be worth getting additional feedback from better writers/editors.A few comments about substance:– The interpretation and necessity of controlling for phylogenetic non-independence. Having read the revised manuscript, your replies and the comments of Reviewer 4, I remain confused about the interpretation of the analysis. For example, I don't understand whether it is about controlling for phylogenetic relationships in errors or in factors that affect the relationship between π and Nc. Moreover, to the extent that it does the latter, what are the assumptions that go into the correction.

I agree this was unclear in the way I presented the results; I have fixed this to make it clear I am dealing with phylogenetic correlation in the residuals.

In the discussion you say that (l. 415-16) "The evidence of high phylogenetic signal found in this study suggests PCMs are needed, in part to avoid spurious results from phylogenetic pseudo-replication." If I understand you correctly, you are suggesting that factors that modify the relationship between π and Nc, such as life history traits, are likely to be more similar in more closely related species, and consequently, the (pi, Nc) points corresponding to closely related species may be considered replication in that they reflect the same processes. If this is what you meant, I think it should be spelled out. Moreover, it would be good to do so early on, in the results, such that the reader can better understand what the PCM analysis is plausibly about.

I added a lot in the Results section on pseudo-replication (lines 197-284) to address this and more clearly explain why this correction is needed, and what it does and does not assume.

In the Results section you say that (l. 220-223): "If the relationship between diversity and population size was free of shared phylogenetic history, \λ = 0 and all the variance could be explained by evolution on the tips; this is analogous to Lynch's conjecture that coalescent times should be free of phylogenetic signal (2011)." Doesn't this contradict the interpretation discussed in the previous paragraph?

I agree this was incorrect and have removed this.

Also, I still don't understand the claim that PCM corrects for "spurious pseudo-replication". Shouldn't the determination whether it is "pseudo-replication" depend on the notion of the "true" relationship that you are trying to estimate? Stated differently, maybe some of the phylogenetic signal arises from similarities in factors that affect the pi-Nc relationship and others that just affect π (e.g., mutation rates), a given notion of the "true" relationship would suggest that you want to correct for the latter but not the former. How can PCMs do that without specifying what it is that you are correcting for?I realize that some of these problems may reflect my ignorance about PCMs, but doubt that the general readership you are aiming is much more knowledgeable.

This gets at some deeper issues about PCMs, but I think the important part is that phylogenetic signal in either the X or Y variable itself is not a violation of the standard regression model, only error in the residuals (Revell, 2010). Even if the true relationship is driven by, say, a mutation rate change along a lineage, sister taxa that inherit this mutation rate along their parent lineage are not independent observations of this effect in a way that an entirely different clade with its own mutation rate change would be (Maddison and FitzJohn, 2014). The phylogenetic mixed-effect model is only looking for correlation in errors and down-weighting the samples as if they were correlated observations. This is a conservative approach to deal with the structure found in the residuals, and importantly, the findings are not qualitatively different than the OLS.

– Relationship between genetic map length and Nc. You note that (l 333-5): "These findings are consistent with both the hypothesis that non-adaptive processes increase genome size in small-Ne species (Lynch and Conery, 2003) which in turn could increase map lengths…". I think the map length is largely explained (e.g., R^2=0.96) by the requirement of having one cross-over per chromosome or chromosomal arm (can't remember which). Specifically, I think that this relationship is much stronger than with genome size, and am not sure whether there is any residual effect of size after controlling for the number of chromosomes.

This is an interesting point, and I do have a figure showing the relationship between chromosome number and map length in the project’s GitHub repository. I’ve run a quick regression of map_length ~ chrom_number + genome_size and find both are significant. However, the direction of causality is unclear and this could be plagued by collider bias. Overall, I think the discussion of this is best left as is (the original sentence was added only to address a reviewer’s concerns in the last round of revisions).

– Discussion about different measures of Ne. I found several points in this section deep and insightful. As you point out, there are different N_e's for different quantities, and comparisons among them informs us about different processes. I think it would be helpful to emphasize that these are different quantities rather than estimators of the same quantity and that Lewontin's paradox is specifically about the one defined by diversity levels.

I have tried to make this clear with “These various estimators capture different summaries of effective population size on shorter timescales than coalescent-based estimators”.

A few general comments about presentation:– You often use several different terms for the same thing, e.g., N_e and expected coalescence rate; explain, solve and resolve; genetic/recombinational map length, heterozygosity/pairwise diversity. This is confusing to readers, who wonder whether there was a reason for using the specific term. Choosing one for each term and sticking with it would be clearer.

I have made these changes (though in some cases allelic heterozygosity, not per basepair diversity was measured by these earlier allozyme studies).

– You sometimes use terms that seem to be private abbreviations, e.g., "strong selection parameters" and "quantifying… paradox".

I have made these changes.

– Perhaps avoid hyphenations as abbreviations, e.g., "low-Nc species" and "across-taxa relationship".

I have kept the low-Nc, high-Nc, etc as this saves a lot of text and makes sentences shorter.

- The discussion would benefit from extensive editing.

I have made the majority of these changes, thank you for your feedback!

Reviewer #2 (Recommendations for the authors):I have mixed feelings about this revision. The author did not follow the reviewers' suggestion of shifting from a research to a review article, and rather tried to reinforce his original results. One problem I'm having is that I still do not see the point of the sophisticated phylogenetic analysis that is presented. It is clear from the existing literature that π has a strong phylogenetic inertia; I guess the interesting question is what causes this inertia; whether λ is 0.4, 0.6 or 0.8 is good to know but not a major achievement, in my opinion. I have a similar comment regarding the report that genetic map tends to be shorter in large census sized species, which is cool but not totally novel, and not a very strong effect – I do not understand why social insects are excluded from the model, by the way. I like much figure 4b, which substantiates Coop's 2016 argument via empirical data analysis – but again the added value is mainly in the illustration; the argument existed already.

I think it is important to point out that Coop (2016) only refuted Corbett-Detig et al’s claim that their estimated impact of linked selection is strong enough to explain Lewontin’s Paradox. Since Corbett-Detig’s work does not have census population sizes, only proxies of population size, we have no concept of the magnitude of the effect to be explained, and whether linked selection can explain it. This can be seen in Coop’s (2016) Figure 1, where the dashed red line is a hypothetical relationship — in my work, by quantifying the π-Nc relationship, we can get a sense of the scale. Finally, my work goes a step further than Coop’s work, showing that not only do Corbett-Detig’s estimates not go far enough in showing linked selection can explain Lewontin’s Paradox, but that no estimates, not even those orders of magnitude larger, can explain Lewontin’s Paradox (see Figure 4-Supplement 4).

Reviewer #3 (Recommendations for the authors):In reviewing the author's response letter and the revised draft, the revisions have certainly streamlined the paper and clarified the author's justification for the analyses.However, I still have some concerns about how the work is motivated at the outset, and how the novelty of the results is explained:In the Introduction, the bottom of page 3 the author says that a limitation of past work is that others do not propose a mechanism by which traits act to constrain diversity within a few orders of magnitude. This seems to be a major motivation for doing a new study, but how does this study tackle this question? Did the author have an a priori reason to expect that a more explicit estimation of Nc would resolve Lewontin's paradox, and if so, why? Could the phylogenetic correction have been expected to have resolved Lewontin's paradox, and if so, why (see below)?

See the response to reviewer #2 above. Also, if the relationship between π and Nc were no longer significant after accounting for phylogeny, and within-clade regressions were similarly non-significant, this would not solve Lewontin’s Paradox. Rather, this would suggest that the relationship is even less dependent on Nc than we expected, and perhaps some macroecological or macroevolutionary process could explain this.

Correcting for the effect of Nc on map length in tandem with the linked selection model fitting does seem to fall into this category, and with this in mind, I think the author could go a long way towards clarifying the novelty of the work by being more explicit about these results. As noted by reviewer 1 in the first round of reviews the 'glass half full' interpretation of this study for the genetic draft model is that the map length correction combined with linked selection model fitting CAN actually go a significant way towards shrinking down the range of diversity expected. While it can't go ALL the way, this is an important and novel result that is more important to clarify than to simply conclude that we are still in the exact same conclusion zone as before this work. The author has gone some way towards using linked selection to explain the variation, and this is worth clarifying, and perhaps quantifying in the discussion.

I have addressed this with the main comments. The issue here is that the findings here are qualitative; they show that linked selection models are incapable of explaining Lewontin’s Paradox under strong selection expected of *D. melanogaster*. Given that it would be erroneous to assume that linked selection in all species is as strong as it is in *D. melanogaster*, I cannot use these results to quantify the impact of linked selection across species.

I still find the motivation for the phylogenetic correction hard to parse in the introduction, and I would suggest front-loading some of the points the author makes in the response about the importance of phylogenetic correction even if coalescent times themselves are not constrained by phylogeny into the intro. Clearly, multiple reviewers found it difficult to understand why phylogenetic correction was needed and wasn't just eroding power, so this is important to front-load.

I have clarified this in the PCM section.

Relatedly, I am still not convinced by the argument that for Lynch's conjecture to be true λ must be zero (page 7). A number of possible interpretations made by the author in the results provide possible mechanisms (like phylogenetic changes in mutation rate) that would still enable coalescent times themselves to be free of a direct phylogenetic effect. The revisions and response now provide a better explanation for the importance of a phylogenetic correction in any case, but I don't think any of the analyses have definitively told us that coalescent times have a phylogenetic signal- it could simply be the phylogenetic inertia of traits associated with Ne and mutation rate.

I discuss this in the discussion a bit (see the mention of Uyeda et al. around lines 432-438). Overall, a trait that has phylogenetic signal is not a violation of the regression model, and not a problem; only if the residuals have correlated errors is this a problem (see Revell 2010 and Uyeda et al. 2018).